# Random intercept and linear mixed models including heteroscedasticity in a logarithmic scale: Correction terms and prediction in the original scale

Ricardo Ramírez-Aldana[1]*, Lizbeth Naranjo[2]

**1** Instituto Nacional de Geriatría, Ciudad de México, Mexico, **2** Departamento de Matemáticas, Facultad de Ciencias, Universidad Nacional Autónoma de México, Ciudad de México, Mexico

* ricardoramirezaldana@gmail.com

**Data Availability Statement:** Data are available from Encuesta Nacional de Ingresos y Gastos de los Hogares, (ENIGH) 2016 (https://www.inegi.org.

## Abstract

Random intercept models are linear mixed models (LMM) including error and intercept random effects. Sometimes heteroscedasticity is included and the response variable is transformed into a logarithmic scale, while inference is required in the original scale; thus, the response variable has a log-normal distribution. Hence, correction terms should be included to predict the response in the original scale. These terms multiply the exponentiated predicted response variable, which subestimates the real values. We derive the correction terms, simulations and real data about the income of elderly are presented to show the importance of using them to obtain more accurate predictions. Generalizations for any LMM are also presented.

## Introduction

In economics and other scientific areas such as medicine, geology, and genetics; it is common to study linear models with a dependent variable defined in a logarithmic scale; for instance in studies related to income [1], health insurance [2], medical expenditures [3], health care utilization and earnings [4], or sediment discharge [5]. In the logarithmic scale, the variable can have an associated normal distribution, whereas in the original scale this is not true. In other words, dependent variables correspond to random variables with a log-normal distribution, a skewed distribution associated with variables taking only positive values, which has been extensively used in analyses for real data corresponding to stock prices, income (without higher-income individuals), time from infection to first symptoms, distribution of particles, number of words per sentence, age of marriage, size of living tissue, etc. There are instances in which presence of heteroscedasticity can be solved considering such logarithmic scale; however, sometimes this issue is not solved even after the transformation, for instance when the variability is not proportional to the squared conditional mean response given values of the explanatory variables. Additionally, there are data in which nesting between observations is present, for instance, when observations belong to the same spatial cluster. In this case,

mx/programas/enigh/nc/2016/). R code to
reproduce the analyses concerning the ENIGH data
sets, including instructions to obtain the analyzed
data from the ENIGH data sets, is available as
supplementary material.

**Funding:** This work was supported by UNAM-
PAPIIT IN118720.

**Competing interests:** The authors have declared
that no competing interests exist.

independence between observations is not satisfied, since the values in a same cluster are cor-
related, and a random intercept model is preferred.

A random intercept model (RIM) in a logarithmic scale is a special type of linear mixed
model (LMM) [6, 7], in which:

$$\log(Y_{ij}) = \mathbf{x}'_{ij}\boldsymbol{\beta} + \gamma_i + \epsilon_{ij}, \tag{1}$$

where $i = 1, \ldots, m, j = 1, \ldots, n_i, m$ is the number of clusters, $n_i$ is the number of observations
in the $i$th cluster, $\log(Y_{ij})$ is the response associated with the $j$th observation in the $i$th cluster,
$\mathbf{x}_{ij} = (x_{ij1}, \ldots, x_{ijp})'$ is a vector of dimension $p$ associated with the $j$th observation in the $i$th clus-
ter corresponding to the $p$ fixed effects given in the regression parameters vector $\boldsymbol{\beta} = (\beta_1, \ldots,
\beta_p)'$. Variable $\gamma_i$ represents an intercept random effect associated with cluster $i$, which allows
to model the relationship among observations for each cluster, it has a normal distribution
$N(0, \sigma_\gamma^2)$, additionally, $\gamma_i$, for $i = 1, \ldots, m$, are independent and identically distributed (i.i.d.).
The random error is $\epsilon_{ij}$, and since heteroscedasticity is assumed in (1), $\epsilon_{ij} \sim N(0, \sigma^2 w_{ij}^{-1})$ i.i.
d., where $w_{ij}^{-1}$ is a known number that allows different variability between observations and
clusters. The terms $w_{ij}^{-1}$ are assumed as known; and for instance, they could be obtained using
the unit size or the ELL method [8, 9]. Under this method, two linear models are fitted, a first
model (beta) is the corresponding marginal model and a second one (alpha) is a model associ-
ated with transformed residuals obtained from the beta model (residuals obtained after delet-
ing the effects associated with the random effects); then, an approximation of the terms $w_{ij}^{-1}$
can be used in the random intercept model. Finally, the random effects $\gamma_i$ and the errors $\epsilon_{ij}$ are
independent.

In many cases, it is necessary to return to the original scale of $Y_{ij}$. Traditionally, this is done
by simply applying an exponential function to the predicted values obtained from the model.
However, this approach does not consider that the random terms involved in the model are
transformed as well, and predictions are subestimated. In some cases, a generalized linear
mixed model (GLMM) [10] with an associated distribution according to the data type could be
used (e.g., gamma, Poisson, etc.). However, sometimes it is preferred to use the normal distri-
bution in the logarithmic scale, when we know the dependent variable has a log-normal distri-
bution (as far as we know, it is not one distribution included in programs that fit GLMM; and,
transforming the dependent variable in a normal GLMM would be similar as what we are
doing), or when other processes depend on such normal RIM. For instance, in small area esti-
mation there are methods based on the RIM, e.g. the empirical best predictor (EBP) method
[11], in which parameters are estimated from a RIM using the sample information; after that,
the conditional distribution of the out-of-sample data given the sample data can be derived
from the normal distribution assumption; the predicted values, simulations, and Monte Carlo
approximations are used to estimate poverty measures at a small area level (for elements in or
outside of the sample), and finally, a parametric bootstrap mean squared error (MSE) estima-
tor is obtained based on the same RIM. Additionally, not all possible distributions are imple-
mented in GLMM and a logarithmic transformation must be used in LMM. In this sense;
recently, [12] proposed a model for data showing skewness at the log scale based on an exten-
sion of a distribution called generalized beta of the second kind, which can be seen as a random
effects model designed for skewed response variables extending the usual log-normal-nested
error model, they also found empirical best predictors for poverty measures in small areas.

Statistical models to correct the logarithmic transformation in linear regression models
have been proposed by different authors, e.g. [5, 13, 14]; other authors have used Bayesian
methods to deal with it, e.g. [15]. Some extensions to address heteroscedasticity in linear
regression models with a logarithmic scale have also been proposed, e.g. [2, 16]. Other authors

have compared the logarithmic transformation in linear models with other type of models in different applications, e.g. [4, 17–21]. Moreover, others have studied the Box-Cox transformation in linear models, e.g. [3, 22–24], being the logarithmic transformation a particular case. Finally, a Box-Cox transformation in LMM has been studied by [23].

In this paper are derived the correction terms that should be used to obtain more accurate predictions in a RIM with heteroscedasticity, when predictions are desired in the original scale. In economics, for instance, this allows a more accurate prediction of income or to improve predictions of measures depending on it, for instance poverty measures. These correction terms multiply the exponentiated predicted values obtained from the RIM, calculating the latter values (without correction) being the usual procedure. These terms are important since they allow to obtain more precise predictions, with a smaller MSE. Since the RIM contains two random terms, the random effect and the error term, two correction terms are obtained. When these correction terms are not included, subestimation occurs. Similar terms have been obtained for linear regression models, but, as far as the authors know, they have not been derived for RIM. These results are relevant, not only when a RIM is fitted in some data, but also when methodology is based on such models, for instance in small area estimation.

The motivation example considers a sample of aging individuals over 60 years old in Mexico, in which some household and socio-demographic measures and income are known. The information is available by state $i = 1, \ldots, m$; for $m = 32$, the number of individuals by state is $n_i$, $Y_{ij}$ is the income by individual $j = 1, \ldots, n_i$ and state $i$, and there are a total of $n = \sum_{i=1}^{m} n_i$ observations. Assume we want to estimate the expected income for these individuals, $E[Y_{ij}]$, according to the available explanatory variables. This process could be useful to estimate income for another set of similar individuals, in which income is not available but the other variables are, for imputation, or in a simulation process, as in small-area estimation which depends on simulating income in out-of-sample individuals. In the framework of linear models, there are several options for this estimation; in some of them, the distributional assumptions are better satisfied in a logarithmic scale, using $\log(Y_{ij})$ as response, but the estimations are required on the original scale. A first option is to estimate the expected income, without a common random effect for state, simply using a linear regression in a logarithmic scale and using a correction term to estimate in the original scale. A second option is fitting a linear model on the log-transformed scale with random effects for state obtaining the exponentiated predicted values to estimate the income. A third option is to associate a gamma distribution to the income, a commonly used distribution for positive skewed data such as the income or costs [25], and fit a generalized linear mixed model including random effects for state. The fourth option we propose is to apply correction terms on the second option. We show here that this improves the precision of the estimated values. We apply the corrections terms to both the real data set and in simulated data to evaluate which of the different described options including random effects has a better performance, for the simulations we variate the number of clusters, observations by cluster, parameters associated with the variance of the random effects and error terms, and consider models under different distributions.

This paper is organized as follows. In the second section, we briefly present the correction term used in a linear regression model in a logarithmic scale, including the so-called smearing estimate. In the third section, we derive correction terms for a RIM with heteroscedasticity, and the corresponding correction terms for a RIM with homoscedasticity are obtained as a particular case. In the fourth section, we obtain simulations using a log-normal distribution to show that the MSE is minimized when the correction terms are used and in certain cases when gamma distribution simulations are used, comparing the estimations using our method with those derived from a generalized linear mixed model. Additionally, the real data

corresponding to income in elderly people is analyzed to show the use of the correction terms and to compare predictions using different options to calculate them. In a fifth section, we propose a generalization for LMM and for transformations different from the logarithm. Finally, the conclusion is presented in the last section, and some of the linear algebra used for the calculations is presented as Supplementary Material.

## Correction term associated with a linear regression model in a logarithmic scale

In our motivation example, assume that we estimate income in a logarithmic scale without considering a random effect for state. In this case, we are in the framework of a linear regression and the second index $j$ is unnecessary, and thus, for simplicity we eliminate it in this section.

A linear regression model in a logarithmic scale, also called log-normal linear model, is defined as:

$$\log(Y_i) = \mathbf{x}_i'\boldsymbol{\beta} + u_i, \tag{2}$$

where $Y_i$ is the response variable for the $i$th observation, $\mathbf{x}_i = (x_{i1}, \ldots, x_{ip})'$ is a vector of the $p$ explanatory variables for the $i$th observation, $\boldsymbol{\beta} = (\beta_1, \ldots, \beta_p)'$ is a vector of dimension $p$ of regression parameters, and $u_i$ is an error term, where $u_i \sim N(0, \sigma^2)$ i.i.d., for $i = 1, \ldots, n$. In matrix notation, $\log(\mathbf{Y}) = \mathbf{X}\boldsymbol{\beta} + \mathbf{u}$, where $\mathbf{Y} = (Y_1, \ldots, Y_n)'$, $\mathbf{X} = (\mathbf{x}_1, \ldots, \mathbf{x}_n)'$, and $\mathbf{u} = (u_1, \ldots, u_n)'$.

From (2), the expected value of the response is $\mathrm{E}[\log(Y_i)] = \mathbf{x}_i'\boldsymbol{\beta}$. However, since $Y_i = \exp(\mathbf{x}_i'\boldsymbol{\beta})\exp(u_i)$, in the original scale, we have that the expected value is $\mathrm{E}[Y_i] = \exp(\mathbf{x}_i'\boldsymbol{\beta})\mathrm{E}[\exp(u_i)]$. Omitting subindex $i$, we have that:

$$
\begin{aligned}
\mathrm{E}[\exp(u)] &= \int \exp(u)\mathrm{d}F_u = \int \exp(u)\frac{1}{\sqrt{2\pi\sigma^2}}\exp\left(-\frac{1}{2\sigma^2}u^2\right)\mathrm{d}u \\
&= \exp(\sigma^4/(2\sigma^2))\int \frac{1}{\sqrt{2\pi\sigma^2}}\exp\left(-\frac{1}{2\sigma^2}(u-\sigma^2)^2\right)\mathrm{d}u.
\end{aligned}
$$

Noticing that the integral in the last equality is equal to one, hence,

$$\mathrm{E}[\exp(u)] = \exp((1/2)\sigma^2), \tag{3}$$

and so $\mathrm{E}[Y_i] = \exp(\mathbf{x}_i'\boldsymbol{\beta})\exp((1/2)\sigma^2)$. Therefore, the estimator of the predicted response is given by

$$\hat{\mathrm{E}}[Y_i] = \exp(\mathbf{x}_i'\hat{\boldsymbol{\beta}})\exp((1/2)\hat{\sigma}^2).$$

Using the same reasoning and considering heteroscedasticity in (2), allowing to have different variability among subjects, i.e. $u_i \sim N(0, \sigma^2 w_i^{-1})$, $\mathrm{E}[Y_i] = \exp(\mathbf{x}_i'\boldsymbol{\beta})\exp((1/2)\sigma^2 w_i^{-1})$.

The last term can be estimated by replacing $\beta$ with the least squares estimator $\hat{\boldsymbol{\beta}} = (\mathbf{X}'\mathbf{X})^{-1}\mathbf{X}'\log(\mathbf{Y})$, and $\sigma^2$ with the biased (maximum likelihood, ML) or unbiased estimator, $\hat{\sigma}^2 = RSS/n$ or $\hat{\sigma}^2 = RSS/(n-p)$, respectively, where $RSS = (\log(\mathbf{Y}) - \mathbf{X}\hat{\boldsymbol{\beta}})'(\log(\mathbf{Y}) - \mathbf{X}\hat{\boldsymbol{\beta}})$ is the residual sum of squares. Observe that $\hat{\mathrm{E}}[Y_i] = \exp(\mathbf{x}_i'\hat{\boldsymbol{\beta}})\exp((1/2)\hat{\sigma}^2 w_i^{-1})$ is the estimator of the predicted response associated with a log-normal distribution.

A modified non-parametric estimator can also be associated with model (2) by using the smearing estimate [3]. We assume $u_i \sim F$ i.i.d., $i = 1, \ldots, n$, where $\mathrm{E}[u_i] = 0$ and $Var(u_i) = \sigma^2$.

Since $F$ is not completely known, the empirical distribution,

$$F_n(u) = \frac{1}{n}\sum_{i=1}^{n} I_{\{\hat{u}_i \leq u\}},$$

is used, where from (2) $\hat{u}_i = \log(Y_i) - \mathbf{x}_i'\hat{\boldsymbol{\beta}}$ is the estimated value of $u_i$, and the indicator function $I_{\{\hat{u}_i \leq u\}}$ is equal to 1 if $\hat{u}_i \leq u$ and 0 otherwise.

Assuming $Y_0$ corresponds to an observed response with associated explanatory variables values $\mathbf{x}_0$, the predicted response is:

$$\mathrm{E}[Y_0] = \int \exp(\mathbf{x}_0'\boldsymbol{\beta} + u)dF_n(u) = \frac{1}{n}\sum_{i=1}^{n}\exp(\mathbf{x}_0'\boldsymbol{\beta} + \hat{u}_i).$$

Furthermore, substituting the regression parameter $\boldsymbol{\beta}$ by the estimates $\hat{\boldsymbol{\beta}}$, the estimated predicted response is given by:

$$\hat{\mathrm{E}}[Y_0] = \exp(\mathbf{x}_0'\hat{\boldsymbol{\beta}})\frac{1}{n}\sum_{i=1}^{n}\exp(\hat{u}_i).$$

## Correction terms in a RIM with heteroscedasticity and a logarithmic scale

In this section we derive the correction terms for a RIM with heteroscedasticity in a logarithmic scale. In terms of our motivation example, the process corresponds to estimate the expected income by fitting a model in a logarithmic scale adding a common random effect for state and using correction terms that allow more precise estimations in the original scale. First, a preliminar estimator is introduced. Second, an estimator based on the random effect best linear predictor is presented. Third, an estimator based on a conditional expectation is proposed. Finally, a correction term based on the smearing estimate is given.

### A preliminar estimator

From the RIM model given in (1), equivalent to $Y_{ij} = \exp(\mathbf{x}_{ij}'\boldsymbol{\beta} + \gamma_i + \epsilon_{ij})$, then, by using independence between $\gamma_i$ and $\epsilon_{ij}$, the expectation of the response in the original scale is:

$$\mathrm{E}[Y_{ij}] = \exp(\mathbf{x}_{ij}'\boldsymbol{\beta})\mathrm{E}[\exp(\gamma_i)]\mathrm{E}[\exp(\epsilon_{ij})]. \tag{4}$$

As in (3), the expectations of the exponentials of $\gamma_i$ and $\epsilon_{ij}$ are $\mathrm{E}[\exp(\gamma_i)] = \exp((1/2)\sigma_\gamma^2)$ and $\mathrm{E}[\exp(\epsilon_{ij})] = \exp((1/2)\sigma^2 w_{ij}^{-1})$, and, as a consequence,

$$\mathrm{E}[Y_{ij}] = \exp(\mathbf{x}_{ij}'\boldsymbol{\beta})\exp((1/2)\sigma_\gamma^2)\exp((1/2)\sigma^2 w_{ij}^{-1}).$$

Therefore, using the corresponding estimators,

$$\hat{\mathrm{E}}[Y_{ij}] = \exp(\mathbf{x}_{ij}'\hat{\boldsymbol{\beta}})\exp((1/2)\hat{\sigma}_\gamma^2)\exp((1/2)\hat{\sigma}^2 w_{ij}^{-1}), \tag{5}$$

where $\hat{\sigma}^2$ and $\hat{\sigma}_\gamma^2$ are variance estimators corresponding to the error and random effects terms, respectively, and $\hat{\boldsymbol{\beta}}$ is the fixed effects estimator, estimated by using ML or restricted ML estimator (REML) methods.

## An estimator based on the random effect best linear predictor

The predicted values in (5) do not consider that the random effect $\gamma_i$ can be estimated through the best linear predictor,

$$\hat{\gamma}_i = \mathrm{E}[\gamma_i|\log(\mathbf{Y})], \tag{6}$$

thus having a predictor for each $i$th observation, for $i = 1, \ldots, m$. The vector of estimated random effects corresponds to $\hat{\gamma} = \mathrm{E}[\gamma|\log(\mathbf{Y})]$.

Similarly, to obtain a better predictor associated with $Y_{ij}$, it is more adequate to use $\mathrm{E}[\exp(\gamma_i)|\log(\mathbf{Y})]$, instead of $\mathrm{E}[\exp(\gamma_i)]$, in (4). Hence, the predictor is:

$$\hat{\mathrm{E}}[Y_{ij}] = \exp(\mathbf{x}'_{ij}\hat{\boldsymbol{\beta}})\hat{\mathrm{E}}[\exp(\gamma_i)|\log(\mathbf{Y})]\hat{\mathrm{E}}[\exp(\epsilon_{ij})]. \tag{7}$$

A first approach to estimate $\mathrm{E}[\exp(\gamma_i)|\log(\mathbf{Y})]$ could be simply by using $\hat{\mathrm{E}}[\exp(\gamma_i)|\log(\mathbf{Y})] = \exp(\hat{\gamma}_i)$, so the estimator (7) would be

$$\hat{\mathrm{E}}[Y_{ij}] = \exp(\mathbf{x}'_{ij}\hat{\boldsymbol{\beta}} + \hat{\gamma}_i)\hat{\mathrm{E}}[\exp(\epsilon_{ij})].$$

Note that,

$$\exp(\mathbf{x}'_{ij}\hat{\boldsymbol{\beta}} + \hat{\gamma}_i) \tag{8}$$

is the predicted value corresponding to $\log(Y_{ij})$ exponentiated to return to the original scale (naive estimator). This term is multiplied by a term associated with the error. Assuming heteroscedasticity, $\hat{\mathrm{E}}[\exp(\epsilon_{ij})] = \exp((1/2)\hat{\sigma}^2 w_{ij}^{-1})$, and the estimator (7) would be:

$$\hat{\mathrm{E}}[Y_{ij}] = \exp(\mathbf{x}'_{ij}\hat{\boldsymbol{\beta}} + \hat{\gamma}_i)\exp((1/2)\hat{\sigma}^2 w_{ij}^{-1}). \tag{9}$$

Note that, according to the Jensen inequality,

$$\exp(\hat{\gamma}_i) = \exp(\mathrm{E}[\gamma_i|\log(\mathbf{Y})]) \le \mathrm{E}[\exp(\gamma_i)|\log(\mathbf{Y})],$$

thus, $\exp(\hat{\gamma}_i)$ subestimates $\mathrm{E}[\exp(\gamma_i)|\log(\mathbf{Y})]$. Hence, a better prediction can be derived by directly obtaining $\mathrm{E}[\exp(\gamma_i)|\log(\mathbf{Y})]$.

## An estimator based on $\mathrm{E}[\exp(\gamma_i)|\log(\mathbf{Y})]$

In this subsection, we obtain a better predictor by computing directly the conditional expectation $\mathrm{E}[\exp(\gamma_i)|\log(\mathbf{Y})]$. For this purpose, first we derived the conditional distribution of the random effect $\gamma_i$ conditional to the transformed response for the sample, $\log(\mathbf{Y}) = (\log(\mathbf{Y}_1), \ldots, \log(\mathbf{Y}_m))'$, which is a vector of dimension $n$, where $n = \sum_{i=1}^m n_i$ is the sample size, with $\mathbf{Y}_i = (Y_{i1}, \ldots, Y_{in_i})$ for $i = 1, \ldots, m$. The random effect has an univariate distribution $\gamma_i \sim N(0, \sigma^2)$, whereas $\log(\mathbf{Y})$ has a multivariate distribution $\log(\mathbf{Y}) \sim N_n(\mathbf{X}\boldsymbol{\beta}, V)$, where $\mathbf{X}$ is the design matrix of dimension $n \times p$ of fixed effects associated with the response, $\boldsymbol{\beta}$ is a vector of dimension $p$ of regression parameters, and $V$ is the variance and covariance matrix $\mathrm{Var}[\log(\mathbf{Y})]$ with dimension $n \times n$. The expected value of this conditional distribution corresponds to the predictor given in (6), whereas using properties concerning the distribution of conditioned multivariate normal random variables, it can be shown (see Proposition 1 in Supplementary Material) that the variance associated with the conditional distribution corresponds to

$$\mathrm{Var}(\gamma_i|\log(\mathbf{Y})) = \sigma_\gamma^2 \left(1 - \frac{\sigma_\gamma^2}{\sigma_\gamma^2 + \frac{\sigma^2}{\sum_{j=1}^{n_i} w_{ij}}}\right). \tag{10}$$

Thus,

$$\gamma_i \mid \log(\mathbf{Y}) \sim N\left(\hat{\gamma}_i, \ \sigma_\gamma^2\left(1 - \frac{\sigma_\gamma^2}{\sigma_\gamma^2 + \frac{\sigma^2}{\sum_{j=1}^{n_i} w_{ij}}}\right)\right).$$

Using the result given in (3), corresponding to the expected value associated with a lognormal random variable,

$$\mathrm{E}[\exp(\gamma_i)|\log(\mathbf{Y})] = \exp(\hat{\gamma}_i)\exp\left((1/2)\sigma_\gamma^2\left(1 - \frac{\sigma_\gamma^2}{\sigma_\gamma^2 + \frac{\sigma^2}{\sum_{j=1}^{n_i} w_{ij}}}\right)\right).$$

As a consequence, the predictor of the response in the original scale, $\hat{\mathrm{E}}[Y_{ij}]$, is estimated considering heteroscedasticity and a predictor $\mathrm{E}[\exp(\gamma_i)|\log(\mathbf{Y})]$, for each $i = 1, \ldots, m$, and corresponds to

$$\exp\left(\mathbf{x}'_{ij}\hat{\boldsymbol{\beta}} + \hat{\gamma}_i\right)\exp\left((1/2)\hat{\sigma}_\gamma^2\left(1 - \frac{\hat{\sigma}_\gamma^2}{\hat{\sigma}_\gamma^2 + \frac{\hat{\sigma}^2}{\sum_{j=1}^{n_i} w_{ij}}}\right)\right)\exp\left((1/2)\hat{\sigma}^2 w_{ij}^{-1}\right). \tag{11}$$

From (11) and assuming $w_{ij}^{-1} = 1$ (or $w_{ij} = 1$), which is a model with homoscedasticity in the error term, $\sum_{j=1}^{n_i} w_{ij} = n_i$, and the predictor $\hat{\mathrm{E}}[Y_{ij}]$ corresponds to

$$\exp\left(\mathbf{x}'_{ij}\hat{\boldsymbol{\beta}} + \hat{\gamma}_i\right)\exp\left((1/2)\hat{\sigma}_\gamma^2\left(1 - \frac{\hat{\sigma}_\gamma^2}{\hat{\sigma}_\gamma^2 + \hat{\sigma}^2/n_i}\right)\right)\exp((1/2)\hat{\sigma}^2). \tag{12}$$

Observe how the predicted values given in (11) or (12) include the term $\exp(\mathbf{x}'_{ij}\hat{\boldsymbol{\beta}} + \hat{\gamma}_i)$, which is the naive estimator associated with $Y_{ij}$. This value is corrected according to two factors, one corresponding to the error and another to the random effect. In contrast, the predictor in (9) only considered the term associated with the error term.

Under heteroscedasticity, the predictor given in (9) subestimates the real value since the term $\mathrm{E}[\exp(\gamma_i)|\log(\mathbf{Y})]$, $i = 1, \ldots, m$, is not used. However, it can be easier to calculate since the sum $\sum_{j=1}^{n_i} w_{ij}$ is not included. Once, $\mathrm{E}[\exp(\gamma_i)|\log(\mathbf{Y})]$ is calculated, the predictor is given in (11). As far as we know, this expected value had not been obtained before.

Observe that all estimators given in (9), (11), and (12) consider that a normal distribution is associated with the transformed data.

## A correction term based on the smearing estimate

We saw in the second section that a smearing estimator [3] is a nonparametric statistic used to estimate the expected response on the untransformed scale after fitting a linear model on the transformed scale, thus being useful when the normality assumption is not satisfied. In this subsection, we used this type of estimator to obtain correction terms for the RIM. One variant of the estimators in a model considering homoscedasticity, Eq (12), is obtained by using a smearing estimate for the error term:

$$\hat{\mathrm{E}}[Y_{ij}] = \exp\left(\mathbf{x}'_{ij}\hat{\boldsymbol{\beta}} + \hat{\gamma}_i\right)\exp\left((1/2)\hat{\sigma}_\gamma^2\left(1 - \frac{\hat{\sigma}_\gamma^2}{\hat{\sigma}_\gamma^2 + \hat{\sigma}^2/n_i}\right)\right)\frac{1}{n}\sum_{i=1}^{m}\sum_{j=1}^{n_i}\exp(\hat{\epsilon}_{ij}). \tag{13}$$

One variant, considering different variance in each $i$th cluster, and the corresponding

smearing estimate, is:

$$\hat{E}[Y_{ij}] = \exp\left(\mathbf{x}_{ij}'\hat{\boldsymbol{\beta}} + \hat{\gamma}_i\right)\exp\left((1/2)\hat{\sigma}_\gamma^2\left(1 - \frac{\hat{\sigma}_\gamma^2}{\hat{\sigma}_\gamma^2 + \hat{\sigma}^2/n_i}\right)\right)\frac{1}{n_i}\sum_{j=1}^{n_i}\exp(\hat{\epsilon}_{ij}).$$

## Experimental results

In this section, the proposed correction terms for RIM with heteroscedasticity in a logarithmic scale are applied to simulation-based scenarios and to an income for elderly people real dataset.

### Simulation-based experiment

A simulation-based experiment is conducted to analyse the correction terms proposed in this paper.

The goal of this simulation experiment is to demonstrate that the proposed approach implementation properly works, and, therefore, the real values are adequately recovered by the estimated ones. We generated one hundred datasets for different scenarios, the generated covariates and general structure are as follows.

A set of $m$ clusters having $n_i$ observations each one, for $i = 1, \ldots, m$, are simulated. For balanced designs $m = \{50, 100\}$ and $n_i = \{10, 20\} \,\forall i$. For unbalanced designs there are two scenarios, one with $n_i = \{11, 12, \ldots, 50\}$ and $m = 40$, and another with $n_i = \{11, 12, \ldots, 90\}$ and $m = 80$. The variables $x_{il}$ are randomly generated from a uniform distribution $U(0, 1)$, for $l = 1, \ldots, p$ and $i = 1, \ldots, n$, where $p = 3$ and $n = \sum_{i=1}^{m} n_i$. In order to include an intercept term, $x_{i1} = 1$. These values are the entries in the design matrix $\mathbf{X}$ of dimension $n \times p$. The regression parameters vector is $\beta = (0.8, 1.3, -0.7)'$.

The intercept random effects $\gamma_i$, for $i = 1, \ldots, m$, are generated from a normal distribution with mean 0 and variance $\sigma_\gamma^2 = \{0.2, 0.4\}$, $\gamma_i \sim N(0, \sigma_\gamma^2)$.

In order to include heteroscedasticity, fixed values were proposed for the weights $w_{ij}$. They have been deterministically assigned as $w_{ij} = (i + 1)/10 + j/1000$, for $i = 1, \ldots, m$ and $j = 1, \ldots, n_i$. The error terms vector $\epsilon$ is generated from a multivariate normal distribution $N_n(0, R)$, where $R = \text{diag}(\Sigma_1, \ldots, \Sigma_m)$, $\Sigma_i = \text{diag}(\sigma^2 w_{i1}^{-1}, \ldots, \sigma^2 w_{in_i}^{-1})$, and $\sigma^2 = \{0.2, 0.4\}$. Note that, by using properties of the multivariate normal distribution, it is also possible to generate $\epsilon$ by the following way: first simulate $\epsilon^*$ from a multivariate standard normal distribution $N_n(\mathbf{0}, I_{n \times n})$, or equivalently generate $\epsilon_{ij}^*$ from a univariate standard normal distribution $N(0, 1)$, then do $\epsilon = R^{1/2} \epsilon^*$ where $R^{1/2}$ is such that $R = R^{1/2} R^{1/2}$, or equivalently $\epsilon_{ij} = \sigma w_{ij}^{-1/2} \epsilon_{ij}^*$.

Finally, the response variable in the logarithmic scale is obtained from (1), this is, by substituting the simulated values in $\log(Y_{ij}) = \mathbf{x}_{ij}'\boldsymbol{\beta} + \gamma_i + \epsilon_{ij}$, thus, the response variable $Y_{ij}$ is obtained as $Y_{ij} = \exp(\log(Y_{ij}))$.

Fig 1 shows one simulated dataset. These graphics show that the response variable $\log(Y)$ has a linear relation with $\mathbf{X}\boldsymbol{\beta}$ (graphic in the left). In contrast, as sometimes occurs in practice, a logarithm transformation is needed on $Y$ to get a linear relationship with the explanatory variables (graphic in the right).

Fig 2 shows the simulated response variable $\log(Y)$ and $Y$, compared with their estimated responses. The squared red dots represent the naive estimates without correction terms in (8), and the blue triangles represent the estimated values obtained by using the correction terms in (11). Note that in general the estimates by using the naive estimator are lower than the

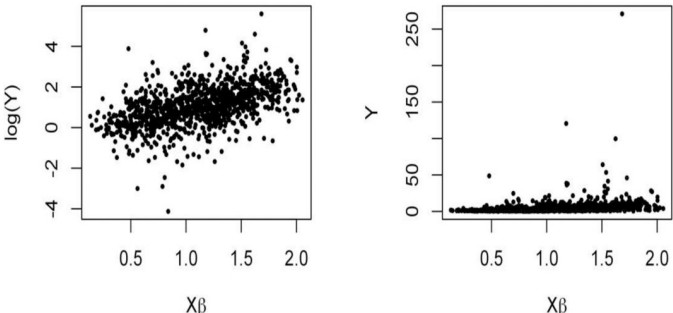

**Fig 1. Simulated data.**

estimates obtained by using the proposed correction terms in (11), showing that the naive estimator subestimates the real values.

Multiple data sets were generated according to the specifications provided in the above paragraphs, and the model's performance was analyzed by using the mean squared error (MSE), given by

$$MSE = \frac{1}{n}\sum_{i=1}^{m}\sum_{j=1}^{n_i}(Y_{ij} - \hat{\mathrm{E}}[Y_{ij}])^2.$$

Table 1 shows the means and standard deviations (sd) associated with the MSE for the one hundred datasets simulated for each scenario defined according to different values of $m$, $n_i$, $\sigma^2$, and $\sigma_\gamma^2$. The MSE are computed by using different estimates $\hat{\mathrm{E}}[Y_{ij}]$, in specific, first by using the naive estimator of (8) (column $MSE_{naive}$), and then by using the correction terms of (5), (9), (11), and (13) (columns $MSE_{(5)}$, $MSE_{(9)}$, $MSE_{(11)}$, and $MSE_{(13)}$ respectively). Finally, a GLMM with a gamma distribution and a logarithmic link is fitted (column $MSE_{Gamma}$), being this an alternative to model positive skewed variables avoiding fitting a transformed response in a LMM.

From these simulation scenarios it is shown that, assuming heteroscedasticity, the best estimations, with the lowest MSE mean and sd, are in general those obtained by using the correction terms given in (11). Standard deviations are always larger in column $MSE_{(9)}$. Moreover, just in one case the means and sd's in column $MSE_{Gamma}$ are lower than others.

Another type of datasets were generated from a GLMM with a gamma distribution associated with the response $Y_{ij}$ and logarithmic link. The values of the parameters are similar to the ones used in the previous simulation experiment concerning the RIM in a logarithmic scale, having analogous balanced and unbalanced designs with the same values of $m$, $n_i$, $x_{il}$, $p$, $\boldsymbol{\beta}$, and $\sigma_\gamma^2$; and including the heteroscedasticity terms $w_{ij}$. The response variable of the GLMM with

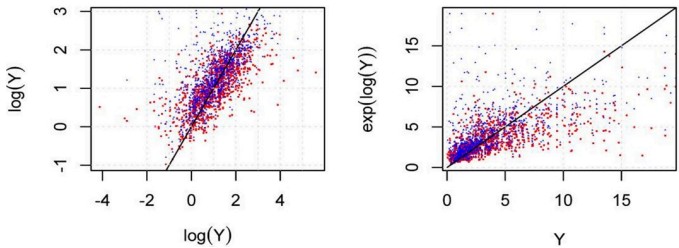

**Fig 2. Simulated response *vs*. estimated response.**

**Table 1. Summary of the MSE for different values of $m$, $n_i$, $\sigma^2$, and $\sigma_\gamma^2$, for data simulated from a RIM in a logarithmic scale.**

| $m$ | $n_i$ | $\sigma^2$ | $\sigma_\gamma^2$ | $MSE_{naive}$ mean (sd) | $MSE_{(5)}$ mean (sd) | $MSE_{(9)}$ mean (sd) | $MSE_{(11)}$ mean (sd) | $MSE_{(13)}$ mean (sd) | $MSE_{Gamma}$ mean (sd) |
|---|---|---|---|---|---|---|---|---|---|
| **Balanced design** | | | | | | | | | |
| 50 | 10 | 0.2 | 0.2 | 4.4 (2.4) | 8.2 (3.3) | 4.0 (2.2) | 4.0 (2.2) | 4.2 (2.3) | 4.6 (2.4) |
| 50 | 10 | 0.2 | 0.4 | 6.4 (4.2) | 16.1 (6.6) | 5.8 (3.6) | 5.8 (3.6) | 6.1 (4.0) | 6.6 (4.2) |
| 50 | 10 | 0.4 | 0.2 | 26.2 (81.7) | 29.5 (80.0) | 24.0 (79.0) | 23.9 (78.8) | 25.3 (81.1) | 25.0 (79.2) |
| 50 | 10 | 0.4 | 0.4 | 171.4 (1233.8) | 181.8 (1221.8) | 166.2 (1221.5) | 165.9 (1220.1) | 169.3 (1228.7) | 160.7 (1163.1) |
| 50 | 20 | 0.2 | 0.2 | 5.4 (5.8) | 9.3 (6.1) | 5.0 (5.4) | 5.0 (5.4) | 5.2 (5.7) | 5.3 (5.6) |
| 50 | 20 | 0.2 | 0.4 | 8.9 (21.2) | 21.2 (36.1) | 8.1 (17.9) | 8.1 (17.9) | 8.6 (20.3) | 8.8 (22.1) |
| 50 | 20 | 0.4 | 0.2 | 19.7 (18.4) | 23.3 (18.0) | 17.5 (16.2) | 17.5 (16.1) | 18.9 (17.9) | 18.1 (16.9) |
| 50 | 20 | 0.4 | 0.4 | 34.1 (61.3) | 45.0 (61.9) | 31.1 (57.4) | 31.1 (57.2) | 33.0 (60.6) | 31.5 (57.2) |
| 100 | 10 | 0.2 | 0.2 | 2.9 (2.3) | 6.7 (2.7) | 2.7 (2.1) | 2.7 (2.1) | 2.8 (2.2) | 3.0 (2.3) |
| 100 | 10 | 0.2 | 0.4 | 4.0 (5.5) | 14.0 (7.3) | 3.7 (4.8) | 3.7 (4.7) | 3.9 (5.4) | 4.1 (5.4) |
| 100 | 10 | 0.4 | 0.2 | 19.0 (67.6) | 22.6 (68.1) | 17.7 (64.3) | 17.6 (64.0) | 18.6 (67.1) | 18.2 (64.5) |
| 100 | 10 | 0.4 | 0.4 | 11.9 (10.3) | 22.0 (11.9) | 10.6 (9.1) | 10.6 (9.0) | 11.4 (10.0) | 11.1 (9.4) |
| 100 | 20 | 0.2 | 0.2 | 2.9 (1.6) | 6.5 (2.0) | 2.7 (1.4) | 2.7 (1.4) | 2.9 (1.5) | 2.9 (1.5) |
| 100 | 20 | 0.2 | 0.4 | 3.7 (2.6) | 13.4 (4.6) | 3.4 (2.3) | 3.4 (2.3) | 3.6 (2.6) | 3.6 (2.5) |
| 100 | 20 | 0.4 | 0.2 | 10.5 (13.2) | 14.0 (13.0) | 9.6 (12.8) | 9.6 (12.8) | 10.3 (13.1) | 9.8 (12.7) |
| 100 | 20 | 0.4 | 0.4 | 27.2 (92.0) | 37.0 (91.6) | 25.3 (90.1) | 25.3 (90.0) | 26.7 (91.7) | 25.4 (87.3) |
| **Unbalanced design** | | | | | | | | | |
| {11,...,50} | | 0.2 | 0.2 | 3.1 (1.6) | 6.7 (2.3) | 2.9 (1.4) | 2.9 (1.4) | 3.0 (1.6) | 3.1 (1.6) |
| {11,...,50} | | 0.2 | 0.4 | 4.9 (3.1) | 15.2 (7.3) | 4.6 (2.8) | 4.6 (2.8) | 4.8 (3.1) | 4.9 (3.0) |
| {11,...,50} | | 0.4 | 0.2 | 14.7 (20.2) | 18.2 (19.3) | 13.5 (18.6) | 13.4 (18.5) | 14.3 (20.0) | 13.9 (19.1) |
| {11,...,50} | | 0.4 | 0.4 | 15.8 (15.3) | 25.7 (15.6) | 14.3 (14.2) | 14.3 (14.1) | 15.2 (15.1) | 14.7 (14.4) |
| {11,...,90} | | 0.2 | 0.2 | 1.4 (0.7) | 5.0 (1.4) | 1.4 (0.6) | 1.4 (0.6) | 1.4 (0.7) | 1.4 (0.7) |
| {11,...,90} | | 0.2 | 0.4 | 2.2 (1.2) | 12.7 (6.2) | 2.1 (1.0) | 2.1 (1.0) | 2.2 (1.2) | 2.2 (1.2) |
| {11,...,90} | | 0.4 | 0.2 | 7.9 (37.5) | 11.6 (37.5) | 7.6 (36.8) | 7.6 (36.8) | 7.8 (37.4) | 7.6 (36.2) |
| {11,...,90} | | 0.4 | 0.4 | 8.9 (15.1) | 18.9 (16.1) | 8.2 (14.3) | 8.2 (14.3) | 8.7 (15.0) | 8.4 (14.2) |

The means and standard deviations (sd) of the MSE are computed by using the estimates, $\hat{E}[Y_{ij}]$, given by the naive estimator, correction terms of (5), (9), (11), and (13) and a GLMM with gamma distribution and logarithmic link, respectively.

gamma distribution and logarithmic link

$$\log(E[Y_{ij}]) = \mathbf{x}_{ij}'\boldsymbol{\beta} + \gamma_i,$$

is thus generated from

$$Y_{ij} \sim Gamma(\texttt{shape} = \alpha w_{ij}, \texttt{scale} = \exp(\mathbf{x}_{ij}'\boldsymbol{\beta} + \gamma_i)/(\alpha w_{ij})),$$

where the probability density function of $Y \sim Gamma(\texttt{shape} = a, \texttt{scale} = s)$ is given by $f_Y(y) = \frac{1}{\Gamma(a)s^a} y^{a-1} e^{-y/s}$, $a > 0$, $s > 0$, and where $\gamma_i \sim N(0, \sigma_\gamma^2)$. The shape parameter $a$ depends on $\alpha$, which was chosen as $\alpha = \{1, 1.5, 5\}$. The purpose of simulating data based on a GLMM with gamma distribution and logarithmic link was to see how our approach worked even when the true distribution associated with the data was not Gaussian. However, our simulations are based on a model extensively used in positive skewed distributions, being this model an alternative to fitting a LMM on the transformed response. In fact, for some particular values assigned to the shape and scale parameters, the distribution associated with the data was similar as that observed for the LMM in the logarithmic scale.

**Table 2. Summary of the MSE for different values of $m$, $n_i$, $\alpha$, and $\sigma^2_\gamma$, for data simulated from a GLMM with gamma distribution and logarithmic link.**

| $m$ | $n_i$ | $\alpha$ | $\sigma^2_\gamma$ | $MSE_{naive}$ mean (sd) | $MSE_{(5)}$ mean (sd) | $MSE_{(9)}$ mean (sd) | $MSE_{(11)}$ mean (sd) | $MSE_{(13)}$ mean (sd) | $MSE_{Gamma}$ mean (sd) |
|---|---|---|---|---|---|---|---|---|---|
| **Balanced design** | | | | | | | | | |
| 50 | 10 | 1 | 0.2 | 13.9 (11.0) | 558.2 (869.0) | 74.6 (74.4) | 90.1 (93.2) | 12.9 (10.7) | 12.0 (10.0) |
| 50 | 10 | 1 | 0.4 | 23.9 (29.2) | 577.4 (895.6) | 51.4 (57.3) | 64.0 (75.6) | 23.7 (28.7) | 19.8 (23.8) |
| 50 | 10 | 1.5 | 0.2 | 9.6 (6.0) | 29.8 (11.3) | 10.7 (5.1) | 11.3 (5.3) | 8.9 (5.7) | 8.5 (5.2) |
| 50 | 10 | 1.5 | 0.4 | 14.0 (9.6) | 40.2 (19.1) | 13.1 (7.9) | 13.7 (8.2) | 13.0 (8.8) | 12.0 (7.9) |
| 50 | 10 | 5 | 0.2 | 2.8 (1.3) | 6.2 (1.8) | 2.6 (1.1) | 2.6 (1.1) | 2.7 (1.2) | 3.0 (1.4) |
| 50 | 10 | 5 | 0.4 | 4.5 (3.4) | 14.3 (7.9) | 4.1 (2.9) | 4.1 (2.9) | 4.3 (3.2) | 4.6 (3.4) |
| 50 | 20 | 1 | 0.2 | 15.2 (8.5) | 311.7 (251.9) | 21.9 (12.7) | 24.0 (14.6) | 16.2 (8.5) | 12.9 (6.7) |
| 50 | 20 | 1 | 0.4 | 26.1 (45.9) | 366.9 (187.1) | 29.5 (35.3) | 31.8 (36.1) | 29.5 (45.5) | 21.8 (36.5) |
| 50 | 20 | 1.5 | 0.2 | 9.0 (2.4) | 24.0 (6.6) | 8.6 (2.0) | 8.7 (2.0) | 8.6 (2.3) | 7.9 (2.0) |
| 50 | 20 | 1.5 | 0.4 | 13.9 (7.3) | 38.4 (14.3) | 12.5 (6.0) | 12.6 (6.0) | 13.7 (6.6) | 12.0 (6.0) |
| 50 | 20 | 5 | 0.2 | 2.7 (1.0) | 6.1 (1.6) | 2.5 (0.9) | 2.5 (0.9) | 2.6 (1.0) | 2.7 (1.0) |
| 50 | 20 | 5 | 0.4 | 4.3 (2.6) | 14.7 (8.1) | 3.9 (2.2) | 3.9 (2.1) | 4.1 (2.4) | 4.2 (2.6) |
| 100 | 10 | 1 | 0.2 | 8.5 (7.8) | 89.5 (54.6) | 16.1 (8.8) | 18.1 (10.1) | 8.1 (7.8) | 7.5 (7.1) |
| 100 | 10 | 1 | 0.4 | 12.2 (7.0) | 110.9 (52.5) | 15.4 (9.1) | 17.3 (11.5) | 11.9 (6.8) | 10.5 (6.0) |
| 100 | 10 | 1.5 | 0.2 | 5.6 (2.7) | 14.9 (4.1) | 5.7 (2.2) | 5.9 (2.2) | 5.3 (2.6) | 5.1 (2.4) |
| 100 | 10 | 1.5 | 0.4 | 7.8 (3.9) | 24.4 (6.9) | 7.3 (3.3) | 7.5 (3.3) | 7.5 (3.7) | 7.0 (3.4) |
| 100 | 10 | 5 | 0.2 | 1.6 (0.5) | 5.2 (1.0) | 1.5 (0.5) | 1.5 (0.4) | 1.6 (0.5) | 1.7 (0.6) |
| 100 | 10 | 5 | 0.4 | 2.4 (1.3) | 12.6 (4.7) | 2.2 (1.1) | 2.2 (1.0) | 2.3 (1.2) | 2.5 (1.4) |
| 100 | 20 | 1 | 0.2 | 8.6 (3.2) | 69.7 (22.0) | 10.0 (4.0) | 10.5 (4.4) | 8.5 (3.3) | 7.5 (2.7) |
| 100 | 20 | 1 | 0.4 | 15.3 (13.5) | 86.0 (28.6) | 14.1 (11.5) | 14.3 (11.5) | 15.8 (13.9) | 13.1 (11.4) |
| 100 | 20 | 1.5 | 0.2 | 5.7 (1.8) | 13.5 (2.2) | 5.2 (1.4) | 5.3 (1.4) | 5.5 (1.7) | 5.0 (1.5) |
| 100 | 20 | 1.5 | 0.4 | 10.3 (10.8) | 25.1 (9.2) | 9.0 (8.9) | 9.1 (8.9) | 10.1 (10.6) | 9.0 (9.1) |
| 100 | 20 | 5 | 0.2 | 1.6 (0.6) | 5.0 (1.0) | 1.5 (0.5) | 1.5 (0.5) | 1.6 (0.6) | 1.6 (0.6) |
| 100 | 20 | 5 | 0.4 | 2.5 (1.3) | 11.8 (4.5) | 2.3 (1.1) | 2.3 (1.1) | 2.4 (1.2) | 2.4 (1.2) |
| **Unbalanced design** | | | | | | | | | |
| {11,. . .,50} | | 1 | 0.2 | 12.6 (10.7) | 75.8 (33.8) | 18.1 (10.9) | 19.8 (11.8) | 11.9 (10.7) | 11.1 (9.7) |
| {11,. . .,50} | | 1 | 0.4 | 17.9 (7.5) | 134.6 (288.9) | 19.4 (9.6) | 21.1 (12.0) | 35.0 (177.4) | 15.2 (6.0) |
| {11,. . .,50} | | 1.5 | 0.2 | 7.7 (2.0) | 15.0 (3.8) | 7.4 (1.6) | 7.6 (1.7) | 7.2 (1.9) | 6.9 (1.7) |
| {11,. . .,50} | | 1.5 | 0.4 | 11.5 (6.6) | 26.6 (14.1) | 10.5 (5.6) | 10.6 (5.6) | 10.8 (6.0) | 10.2 (5.8) |
| {11,. . .,50} | | 5 | 0.2 | 2.2 (0.6) | 5.7 (2.4) | 2.1 (0.5) | 2.1 (0.5) | 2.1 (0.6) | 2.2 (0.6) |
| {11,. . .,50} | | 5 | 0.4 | 3.2 (1.1) | 12.0 (5.6) | 3.0 (1.1) | 3.0 (1.1) | 3.1 (1.1) | 3.2 (1.1) |
| {11,. . .,90} | | 1 | 0.2 | 5.9 (1.4) | 18.5 (4.4) | 6.1 (1.4) | 6.3 (1.5) | 5.7 (1.4) | 5.4 (1.2) |
| {11,. . .,90} | | 1 | 0.4 | 8.0 (2.1) | 28.3 (8.8) | 7.8 (2.1) | 7.9 (2.2) | 7.8 (2.0) | 7.3 (1.9) |
| {11,. . .,90} | | 1.5 | 0.2 | 3.7 (0.7) | 8.0 (1.6) | 3.5 (0.6) | 3.6 (0.6) | 3.6 (0.7) | 3.4 (0.7) |
| {11,. . .,90} | | 1.5 | 0.4 | 6.0 (2.1) | 17.1 (6.3) | 5.5 (1.7) | 5.5 (1.7) | 5.7 (2.0) | 5.5 (1.8) |
| {11,. . .,90} | | 5 | 0.2 | 1.1 (0.2) | 4.5 (1.3) | 1.0 (0.2) | 1.0 (0.2) | 1.1 (0.2) | 1.1 (0.2) |
| {11,. . .,90} | | 5 | 0.4 | 1.6 (0.5) | 11.3 (5.3) | 1.5 (0.4) | 1.5 (0.4) | 1.6 (0.4) | 1.6 (0.5) |

The means and standard deviations (sd) of the MSE are computed by using the estimates, $\hat{E}[Y_{ij}]$, given by the naive estimator, correction terms of (5), (9), (11), and (13), and a GLMM with gamma distribution and logarithmic link, respectively.

Table 2 shows the means and standard deviations (sd) associated with the MSE for the one hundred datasets simulated for each scenario, each one defined according to different values of $m$, $n_i$, $\alpha$, and $\sigma^2_\gamma$; and assuming heteroscedasticity. From these scenarios, it is shown that when the parameter associated with shape $\alpha$ is much bigger than 1, the best estimations, those

having the lowest MSE mean and sd, are in general those obtained by using the correction terms given in (11). Hence, in this case, the estimations by using the RIM in a logarithmic scale and the corrections terms are good, even better than those obtained using a GLMM with a gamma distribution and logarithmic link. However, when $\alpha$ is close to 1 the estimations obtained by using the RIM in a logarithmic scale are worst, which makes sense, since a gamma distribution with parameter $\alpha = 1$ is an exponential distribution, which completely differs from a log-normal distribution.

### Income for elderly people data application

Returning to our motivation example, we performed analyses based on the National Household Income and Expenditure Survey (*Encuesta Nacional de Ingresos y Gastos de los Hogares*, ENIGH) 2016 [26], a biennial study to examine income and its distribution in Mexico. Elderly people were considered (60 or more years old). Quaterly total income, that is the income considering all possible sources of income, was obtained for each person as a response variable. Household and sociodemographic information was considered as well. To avoid presence of outliers, only people with an income between 2,000 and 40,000 Mexican pesos were considered. Hence, a total of $n = 18,512$ participants were included in the analyses.

As already mentioned in the Introduction section, a logarithmic scale was used for the response variable. To help deciding which variables to use as explanatory, we first fitted linear regression models. According to the obtained results, some variables were modified (categories collapsed) or generated using information from other questions. The final linear model in which we are based upon has a coefficient of determination of 0.35. The sociodemographic explanatory variables included in the RIM are: sex, indigeneous (1 = Yes, 2 = No), knowing how to read and write a note (1 = Yes, 2 = No), level of education (0 = None to 9 = Ph.D.), marital status (0 = Without a partner, 1 = With a partner), having a health service provider (1 = Yes, 2 = No), work (1 = Looking for a job, 2 = Retired, 3 = Domestic chores, 4 = Other situation, 5 = Can not work, 6 = Working), disability (0 = Without, 1 = With), and contribution to social security in all their lives (1 = Yes, 2 = No). At a household level, explanatory variables are: number of rooms, presence of wc (1 = Yes, 2 = No), number of light bulbs, household ownership (1 = Rented, 2 = Borrowed, 3 = Owner but paying it, 4 = Owner, 5 = Intestated, 6 = Another situation), number of residents, type of the location where the household is in (0 = Rural, 1 = Urban, a location is considered as urban when its size is of 2,500 or more residents), socioeconomic stratum (1 = Low, 2 = Low medium, 3 = High medium, 4 = High), and flooring material (1 = Ground, 2 = Cement, 3 = Wood, mosaic, or another floor recovering).

Since individuals are nested in each of the 32 states, an intercept random effect for state was included, each state having between 400 and 1000 observations. The parameter (fixed effects) estimations associated with the RIM model with homoscedasticity in the error term are shown in Table 3. The estimated standard deviation associated with the random effect, $\hat{\sigma}_\gamma$, is approximately 0.08, and the corresponding value associated with the error term, $\hat{\sigma}$, is approximately 0.6. A likelihood ratio test comparing the RIM model with a model without the random effect, i.e. $\sigma_\gamma^2 = 0$, was obtained, with an associated $p$-value of less than 0.05 (this number when divided by two is even smaller, a calculation that must be made since the hypothesis involves a value in the frontier of the parametral space). Hence, a random effect is necessary and a linear regression model (without random effects) should not be fitted, which we defined as a first option to possibly use for this data in the Introduction section.

Fig 3 shows the histogram and qq-plot associated with the residuals. They are indicative that the normality assumption is satisfied, the same being true when the random effects qq-plot is examined.

**Table 3. Parameter estimations for the RIM associated with income in a logarithmic scale for elderly people data in 2016.**

| Variable | Value | Std. Error | DF | t-value | p-value |
|---|---|---|---|---|---|
| Intercept | 9.128 | 0.043 | 18261 | 211.533 | <0.001 |
| **Sociodemographic variables** | | | | | |
| Woman | -0.176 | 0.011 | 18261 | -15.476 | <0.001 |
| No indigeneous | 0.042 | 0.010 | 18261 | 4.002 | <0.001 |
| Not knowing how to write/read | -0.064 | 0.017 | 18261 | -3.881 | <0.001 |
| Level of education: Prescholar | -0.069 | 0.106 | 18261 | -0.654 | 0.513 |
| Level of education: Elementary | 0.047 | 0.016 | 18261 | 2.924 | 0.004 |
| Level of education: Junior high | 0.171 | 0.021 | 18261 | 8.002 | <0.001 |
| Level of education: High school | 0.347 | 0.031 | 18261 | 11.088 | <0.001 |
| Level of education: Teacher's school | 0.760 | 0.047 | 18261 | 16.087 | <0.001 |
| Level of education: Technician | 0.222 | 0.028 | 18261 | 7.837 | <0.001 |
| Level of education: Bachelor's degree | 0.443 | 0.028 | 18261 | 15.580 | <0.001 |
| Level of education: Master's degree | 0.625 | 0.079 | 18261 | 7.949 | <0.001 |
| Level of education: Ph.D. | 0.685 | 0.181 | 18261 | 3.793 | <0.001 |
| With a partner | -0.057 | 0.010 | 18261 | -5.682 | <0.001 |
| No health service provider | -0.160 | 0.011 | 18261 | -14.329 | <0.001 |
| Work: Looking for a job | -0.465 | 0.052 | 18261 | -8.989 | <0.001 |
| Work: Retired | -0.120 | 0.013 | 18261 | -9.005 | <0.001 |
| Work: Domestic chores | -0.415 | 0.014 | 18261 | -30.620 | <0.001 |
| Work: Other situation | -0.452 | 0.023 | 18261 | -19.800 | <0.001 |
| Work: Can not work | -0.407 | 0.024 | 18261 | -17.014 | <0.001 |
| With disability | -0.075 | 0.010 | 18261 | -7.430 | <0.001 |
| No contribution social security | -0.201 | 0.012 | 18261 | -17.043 | <0.001 |
| **Household level variables** | | | | | |
| Number of rooms | 0.023 | 0.004 | 18261 | 6.166 | <0.001 |
| No wc | -0.106 | 0.029 | 18261 | -3.578 | <0.001 |
| Total number of light bulbs | 0.014 | 0.001 | 18261 | 10.972 | <0.001 |
| Ownership: Borrowed | -0.130 | 0.026 | 18261 | -4.922 | <0.001 |
| Ownership: Owner but paying | -0.089 | 0.034 | 18261 | -2.611 | 0.009 |
| Ownership: Owner | -0.092 | 0.023 | 18261 | -4.023 | <0.001 |
| Ownership: Intestated | -0.136 | 0.039 | 18261 | -3.525 | <0.001 |
| Ownership: Another situation | -0.115 | 0.059 | 18261 | -1.961 | 0.050 |
| Number of residents | -0.014 | 0.002 | 18261 | -6.271 | <0.001 |
| Urban | -0.005 | 0.012 | 18261 | -0.402 | 0.688 |
| Stratum: Low-medium | 0.096 | 0.014 | 18261 | 6.810 | <0.001 |
| Stratum: High-medium | 0.074 | 0.020 | 18261 | 3.693 | <0.001 |
| Stratum: High | 0.144 | 0.029 | 18261 | 5.034 | <0.001 |
| Floor: Cement | 0.085 | 0.026 | 18261 | 3.222 | 0.001 |
| Floor: Wood, mosaic or other | 0.183 | 0.028 | 18261 | 6.497 | <0.001 |

Fig 4 shows the fitted values for the RIM associated with income in a logarithmic scale for the elderly people data in 2016. The squared red dots represent the naive estimates without correction terms, and blue triangles represent the estimated values by using the correction terms in (12), a particular case of (11), and which, according to the simulation results, are the best estimations (with lowest MSE). Note that the estimates derived through the naive estimator are in general lower than those derived through the proposed correction terms in (12), showing that the naive estimator subestimates the data. In terms of the options discussed in

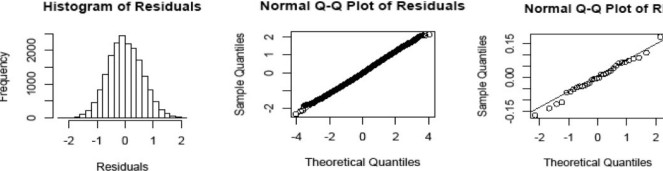

**Fig 3. Residuals.** Left: Histogram of the residuals. Middle: qq-plot of the residuals. Right: qq-plot of the residuals of the random effects.

the Introduction section, the naive estimates and those including the correction terms corresponded to the second and fourth, respectively. When the naive estimator is obtained and compared with the true values, the squared root of the mean squared error is 7027.784, whereas using the correction factor given in (12), the squared root of the mean squared error is 6829.003, which is an improvement. In terms of the third option discussed in the Introduction, we fitted a GLMM using a gamma distribution for the response variable, a logarithmic link function, and both a penalised quasi-likelihood (PQL) and Laplace approximation methods, we checked that the normality assumption in the estimated random effects is satisfied. We obtained values for the squared root of the mean squared error of 6973.41 and 6979.769 under

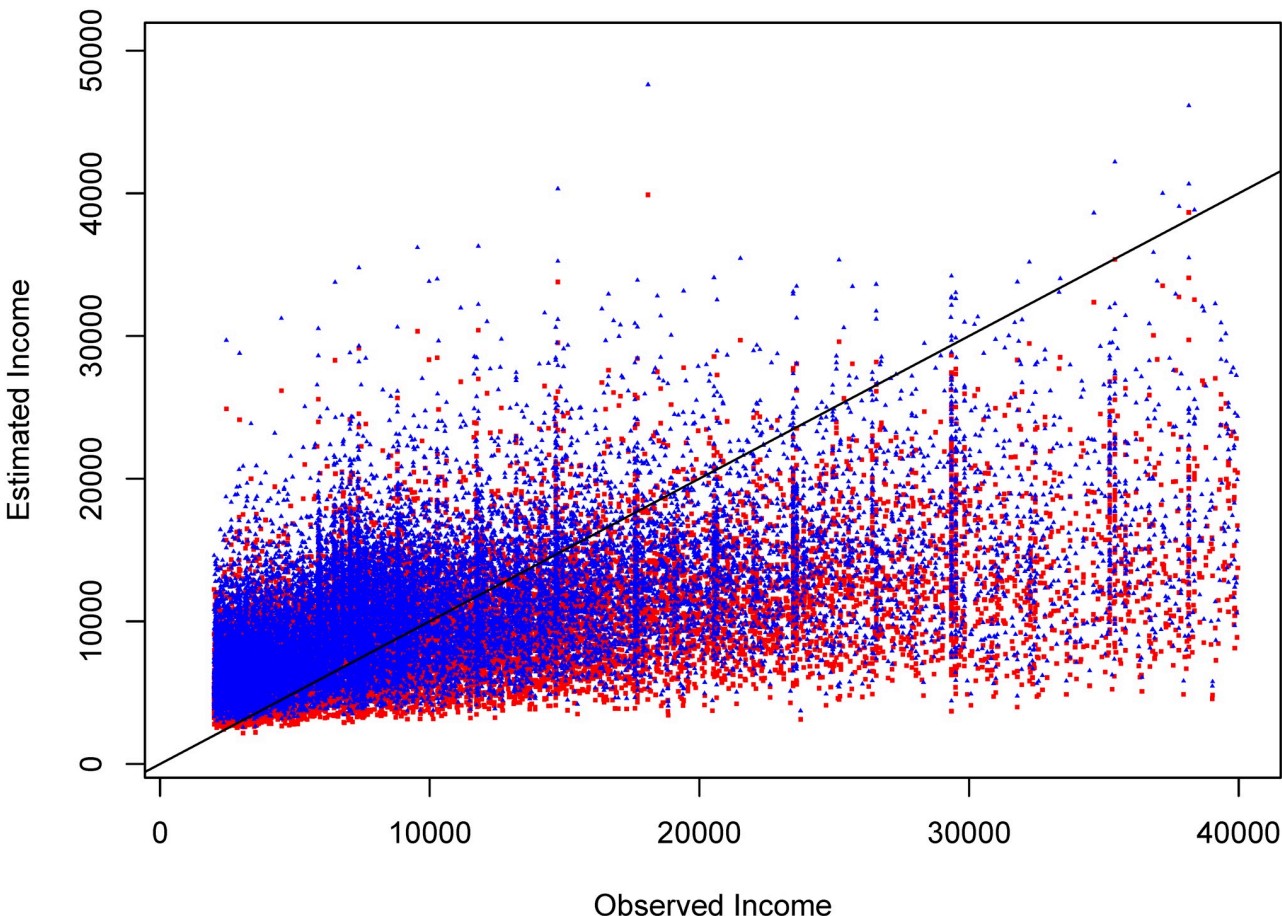

**Fig 4. Fitted values for the RIM associated with income in a logarithmic scale for elderly people data in 2016.** Squared red: naive estimates. Blue triangles: estimates by using the correction terms in (12).

the PQL and Laplace methods, respectively. Hence, in this example the estimates under the correction term are even more precise that those obtained using a GLMM. We fitted models considering some heteroscedasticity schemes, for instance using the ELL method, the cluster size, or the squared residuals, but only with the former method we obtained an inferior mean squared error than under the homoscedasticity scheme; however, the normality assumption was not satisfied as well.

## Mimic simulation example

Validating our proposed correction terms for RIM including heteroscedasticity in a logarithm scale, we did a simulation experiment based on 100 data sets of size 2000. The simulated data approximately mimic the motivating data of the income for elderly people, assuming two types of weights associated with heteroscedasticity: the cluster size and one of the explanatory variables, as sometimes is found in real data. Details are given in the S2 Text.

Our simulation strategy generated the means and standard deviations of the MSE for each one of the corrections terms considering the two types of weights and varying values associated with the variance of the random effects and error terms, see Table 4. The estimations with the lowest MSE corresponded to those obtained using the correction terms associated with Eqs (9) and (11). See details and a table including more values in the Supplementary Material.

## Generalization to linear mixed models and with functions different from the logarithm

In this section, we generalize the correction terms for any LMM and for transformations different from the logarithm. We have seen that the estimators based on the conditional expectancy associated with the random effects given the transformed response have a better performance; thus, we present only this type of estimator for LMM, obtaining a closed

**Table 4. Summary of the MSE for the mimic simulation example.**

| Weights | $\sigma$ | $\sigma_\gamma$ | $MSE_{naive}$ mean (sd) | $MSE_{(5)}$ mean (sd) | $MSE_{(9)}$ mean (sd) | $MSE_{(11)}$ mean (sd) | $MSE_{(13)}$ mean (sd) |
|---|---|---|---|---|---|---|---|
| (1) | 0.15 | 0.079 | 183.0 (5.9) | 797.6 (104.2) | 183.0 (5.9) | 183.0 (5.9) | 183.0 (5.9) |
| (1) | 0.15 | 0.32 | 201.7 (16.2) | 3383.0 (607.6) | 201.7 (16.2) | 201.7 (16.2) | 201.7 (16.2) |
| (1) | 0.15 | 1.28 | 753.6 (350.0) | 35308.6 (19790.8) | 753.8 (350.0) | 753.8 (350.0) | 753.7 (350.0) |
| (1) | 0.594 | 0.079 | 725.3 (22.4) | 1053.2 (84.1) | 724.6 (22.3) | 724.6 (22.3) | 724.7 (22.3) |
| (1) | 0.594 | 0.32 | 796.7 (67.6) | 3432.5 (696.9) | 796.0 (67.5) | 796.0 (67.5) | 796.0 (67.5) |
| (1) | 0.594 | 1.28 | 2994.2 (2264.1) | 34012.2 (27261.6) | 2991.4 (2262.3) | 2991.3 (2262.3) | 2991.9 (2262.5) |
| (1) | 1.2 | 0.079 | 1495.4 (49.6) | 1680.7 (68.5) | 1490.5 (49.1) | 1490.4 (49.1) | 1490.7 (49.1) |
| (1) | 1.2 | 0.32 | 1634.4 (107.4) | 3713.1 (592.2) | 1629.5 (106.4) | 1629.5 (106.4) | 1629.9 (106.5) |
| (1) | 1.2 | 1.28 | 6852.0 (5789.4) | 38593.9 (36263.5) | 6840.1 (5851.4) | 6840.2 (5852.4) | 6844.1 (5869.6) |
| (2) | 0.15 | 0.079 | 540.1 (12.1) | 928.9 (88.2) | 539.9 (12.1) | 539.9 (12.1) | 539.9 (12.1) |
| (2) | 0.15 | 0.32 | 592.8 (38.2) | 3306.8 (562.7) | 592.5 (38.2) | 592.5 (38.2) | 592.6 (38.2) |
| (2) | 0.15 | 1.28 | 2290.2 (1369.8) | 35322.8 (24328.5) | 2288.1 (1367.9) | 2288.1 (1367.9) | 2289.4 (1369.4) |
| (2) | 0.594 | 0.079 | 2264.4 (81.6) | 2387.4 (86.0) | 2245.5 (78.5) | 2245.5 (78.5) | 2251.4 (80.5) |
| (2) | 0.594 | 0.32 | 2506.9 (206.4) | 4311.0 (683.5) | 2484.7 (200.4) | 2484.7 (200.3) | 2491.3 (203.9) |
| (2) | 0.594 | 1.28 | 8994.3 (4847.5) | 34063.3 (21229.0) | 8909.1 (4786.8) | 8908.9 (4786.4) | 8934.1 (4792.5) |
| (2) | 1.2 | 0.079 | 5546.3 (647.1) | 5472.5 (622.3) | 5412.8 (629.8) | 5412.8 (629.7) | 5453.8 (648.7) |
| (2) | 1.2 | 0.32 | 6289.4 (1296.7) | 7195.0 (1295.5) | 6122.4 (1257.0) | 6122.4 (1256.9) | 6182.5 (1301.5) |
| (2) | 1.2 | 1.28 | 23639.7 (17407.2) | 47081.2 (37670.3) | 23141.8 (17392.3) | 23143.1 (17395.0) | 23264.0 (17244.1) |

(1) Size of each cluster. (2) Total number of light bulbs.

formula. A LMM includes $q$ random effects; for instance, we can have random effects associated with some or all the fixed effects. In its matrix form, a LMM corresponds to

$$\log(\mathbf{Y}) = \mathbf{X}\boldsymbol{\beta} + U\boldsymbol{\gamma} + \epsilon,$$

where $\mathbf{X}$ is the design matrix associated with the fixed effects of dimension $n \times p$ and $\boldsymbol{\beta}$ is the corresponding vector of parameters of dimension $p$. On the other hand, $\boldsymbol{\gamma} = (\boldsymbol{\gamma}_1, \ldots, \boldsymbol{\gamma}_m)$ is a vector of dimension $mq$ of random effects, where $\boldsymbol{\gamma}_i$ a vector of dimension $q$ corresponding to all random effects associated with a cluster $i$, with distribution $\boldsymbol{\gamma} \sim N_{mq}(\mathbf{0}, G)$, with $G$ a diagonal matrix of dimension $mq \times mq$, $G = \text{diag}(D, D, \ldots, D)$, where $D$ is the variance and covariance matrix of dimension $q \times q$ associated with the random effects, which is assumed to be the same for all clusters. This term is multiplied by the matrix $U$, a block diagonal matrix of dimension $n \times mq$ given by $U = \text{diag}(U_1, U_2, \ldots, U_m)$, with $U_i$ of dimension $n_i \times q$. The vector of errors has distribution $\epsilon \sim N_n(\mathbf{0}, R)$, where $R$ is a block diagonal matrix of dimension $n \times n$ given by $R = \text{diag}(\Sigma_1, \Sigma_2, \ldots, \Sigma_m)$, with $\Sigma_i$ a diagonal matrix of dimension $n_i \times n_i$ given by $\Sigma_i = \text{diag}(\sigma^2 w_{i1}^{-1}, \sigma^2 w_{i2}^{-1}, \ldots, \sigma^2 w_{in_i}^{-1})$. The error terms and random effects are assumed independent. Considering an individual $j$ in a cluster $i$; $i = 1, \ldots, m$ and $j = 1, \ldots, n_i$, the expression analogous to (1) associated with a LMM is:

$$\log(Y_{ij}) = \mathbf{x}'_{ij}\boldsymbol{\beta} + \mathbf{u}'_{ij}\gamma_i + \epsilon_{ij}, \tag{14}$$

where $\mathbf{u}_{ij}$ is the $j$th row corresponding to matrix $U_i$.

From the joint distribution of the random effects $\boldsymbol{\gamma}$ and transformed response $\log(\mathbf{Y})$, we obtain (see Proposition 2 in S1 File) that the variance and covariance matrix associated with cluster $i$, $\text{Var}(\gamma_i | \log(\mathbf{Y}))$, for $i = 1, \ldots, m$, is

$$\text{Var}(\gamma_i | \log(\mathbf{Y})) = D - DU'_i(U_iDU'_i + \Sigma_i)^{-1}U_iD \tag{15}$$

and

$$\gamma_i \mid \log(\mathbf{Y}) \sim N_q(\hat{\gamma}_i ~,~ \text{Var}[\gamma_i | \log(\mathbf{Y})]),$$

where $\hat{\gamma}_i$ is the best linear predictor of $\gamma_i$, $\hat{\gamma}_i = \text{E}[\gamma_i | \log(\mathbf{Y})]$. Consequently,

$$\mathbf{u}'_{ij}\gamma_i \mid \log(\mathbf{Y}) \sim N(\mathbf{u}'_{ij}\hat{\gamma}_i ~,~ \mathbf{u}'_{ij}\text{Var}[\gamma_i | \log(\mathbf{Y})]\mathbf{u}_{ij}),$$

and using the expected value corresponding to a log-normal distribution:

$$\text{E}[\exp(\mathbf{u}'_{ij}\gamma_i) | \log(\mathbf{Y})] = \exp(\mathbf{u}'_{ij}\hat{\gamma}_i)\exp((1/2)\mathbf{u}'_{ij}\text{Var}[\gamma_i | \log(\mathbf{Y})]\mathbf{u}_{ij}). \tag{16}$$

Thus, to estimate $\text{E}[Y_{ij}]$ in a cluster $i$; $i = 1, \ldots, m$, for an individual $j$; $j = 1, \ldots, n_i$, where $Y_{ij}$ is modeled as in (14), we use the estimator $\exp((1/2)\hat{\sigma}^2 w_{ij}^{-1})$ for the random error $\epsilon_{ij}$, multiplied by the expected value associated with the random effects conditional to the response $\text{E}[\exp(\mathbf{u}'_{ij}\gamma_i) | \log(\mathbf{Y})]$ calculated in (16), and the constant part $\exp(\mathbf{x}'_{ij}\hat{\boldsymbol{\beta}})$. The estimator corresponds to:

$$\exp(\mathbf{x}'_{ij}\hat{\boldsymbol{\beta}} + \mathbf{u}'_{ij}\hat{\gamma}_i)\exp((1/2)\mathbf{u}'_{ij}\text{Var}[\gamma_i | \log(\mathbf{Y})]\mathbf{u}_{ij})\exp((1/2)\hat{\sigma}^2 w_{ij}^{-1}). \tag{17}$$

In (17), all terms are known once substituting the estimated variance and covariance terms for the random effects in $D$ and $\hat{\sigma}^2$ in $\Sigma_i$, both $D$ and $\Sigma_i$ part of $\text{Var}(\gamma_i | \log(\mathbf{Y}))$. These terms and obtained after fitting the model.

For instance, consider a model including random effects associated with the intercept and a variable $u$. For each cluster $i = 1, \ldots, m$, $\gamma_i = (\gamma_{i1}, \gamma_{i2})'$, with $\gamma_{i1}$ and $\gamma_{i2}$ scalars corresponding to

the random effects for the intercept and variable $u$, respectively. The values associated with variable $u$ in cluster $i$ can be accommodated in a vectorial form as $\mathbf{u}_i = (u_{i1}, \ldots, u_{in_i})'$, thus $U_i$ is a matrix of dimension $n_i \times 2$ such that $U_i = (\mathbf{1}_{n_i}, \mathbf{u}_i)'$, where $\mathbf{1}_{n_i}$ corresponds to the intercept. Finally,

$$D = \begin{pmatrix} \sigma^2_{\gamma_1} & \sigma^2_{\gamma_1\gamma_2} \\ \sigma^2_{\gamma_1\gamma_2} & \sigma^2_{\gamma_2} \end{pmatrix}, \tag{18}$$

where $\sigma^2_{\gamma_1}$ and $\sigma^2_{\gamma_2}$ correspond to the variances associated with the random effects for the intercept and variable $u$, respectively, and $\sigma^2_{\gamma_1\gamma_2}$ is the corresponding covariance. It is easy to derive that in this case (15) corresponds to

$$\mathrm{Var}[\gamma_i | \log(\mathbf{Y})] = D - A_i[\sigma^2_{\gamma_1}\mathbf{1}_{n_i}\mathbf{1}'_{n_i} + \sigma^2_{\gamma_1\gamma_2}\mathbf{u}_i\mathbf{1}'_{n_i} + \sigma^2_{\gamma_1\gamma_2}\mathbf{1}_{n_i}\mathbf{u}'_i + \sigma^2_{\gamma_2}\mathbf{u}_i\mathbf{u}'_i + \Sigma_i]^{-1}A'_i,$$

with $A'_i = (\sigma^2_{\gamma_1}\mathbf{1}_{n_i} + \sigma^2_{\gamma_1\gamma_2}\mathbf{u}_i , \ \sigma^2_{\gamma_1\gamma_2}\mathbf{1}_{n_i} + \sigma^2_{\gamma_2}\mathbf{u}_i)$ and $D$ given in (18). This equation can be substituted in expression (17) using estimations of $\sigma^2_{\gamma_1}$, $\sigma^2_{\gamma_2}$, and $\sigma^2_{\gamma_1\gamma_2}$, values obtained after fitting the LMM in any statistical software.

We could consider a transformation more general than a logarithm, for instance a Box-Cox transformation $g$, whose inverse follows a power-normal distribution. Each observation $Y_{ij}$; for $j = 1, \ldots, n_i$ and $i = 1, \ldots, m$, associated with a MLM under a Box-Cox transformation with parameter $\lambda$, $g(Y_{ij})$, satisfies that $g(Y_{ij}) \sim N(\mu, \sigma^2_*)$ with $\mu = \mathbf{x}'_{ij}\boldsymbol{\beta}$ and $\sigma^2_* = \mathbf{u}'_{ij}D\mathbf{u}_{ij} + \sigma^2 w^{-1}_{ij}$. The expected value $\mathrm{E}[X]$ of a power-normal distribution, in this case $X \sim PN(\lambda, \mu, \sigma^2_*)$, is calculated in [27] (Lemma 1). After considering the estimated parameters, this expression corresponds to one class of corrected predictions in the original scale, that without conditioning the random effects to the sample. For instance, for $\lambda = 0$, the expected value given in [27] is $\mathrm{E}(X) = \exp(\mu + \sigma^2_*/2)$, corresponding to Eq (5) when only one random effect is used.

For an invertible function $g(\cdot)$, and considering estimators based on conditioning on the sample, as in (11) for a RIM or (17) for any LMM, a simulation can be used. Assuming that in the transformed scale all normality assumptions are satisfied, we can apply similar results as when a MLM and logarithm transformation were considered, and

$$\mathbf{u}'_{ij}\gamma_i \mid g(\mathbf{Y}) \sim N(\mathbf{u}'_{ij}\hat{\gamma}_i , \ \mathbf{u}'_{ij}\mathrm{Var}[\gamma_i | g(\mathbf{Y})]\mathbf{u}_{ij}), \tag{19}$$

where $\mathrm{Var}[\gamma_i | g(\mathbf{Y})]$ corresponds to (15). The expected value of the response in the original scale in a cluster $i$ for an individual $j$

$$\mathrm{E}[g^{-1}(\mathbf{x}'_{ij}\boldsymbol{\beta} + \mathbf{u}'_{ij}\gamma_i + \epsilon_{ij})|g(\mathbf{Y})].$$

can be approximated with simulations by generating a set of random numbers $z_l$, for $l = 1, \ldots, L$, according to the distribution given in (19), and obtaining:

$$\frac{\sum_{l=1}^{L} g^{-1}(\mathbf{x}'_{ij}\boldsymbol{\beta} + z_l + \epsilon_{ij})}{L},$$

using $\hat{\epsilon}_{ij}$ or $\mathrm{E}[\epsilon_{ij}]$ instead of $\epsilon_{ij}$, the expected value $\mathrm{E}[\epsilon_{ij}]$ could be obtained by simulating the distribution of $\epsilon_{ij}$.

## Conclusion

The correction terms we proposed for a RIM with or without heteroscedasticity with response in a logarithmic scale enable more precise predictions. This is useful since responses in a

logarithmic scale are commonly used, specially in financial and poverty analyses, and with our procedure, we can obtain more precise predictions of an economic measure in a population or better simulations of the distribution of the response, or an associated measure, for a new population (by simulating the error term and random effects and using the values of the explanatory variables). As the simulations assuming log-normal distributions and real data show, the best predictions, with lowest MSE, correspond to those including two correction terms, one for the errors and another for the random effects. These correction terms are easy to calculate and implement without the need of special software.

Even though in a GLMM, a distribution different from the normal can be used, it is sometimes desired simply to work in a logarithmic scale when the normal behaviour under this transformation is properly satisfied; or in other words, when a lognormal distribution adequately fits some data. Besides, through simulations with gamma distributions, a commonly used distribution used to model income or similar variables, we showed that the predictions using the two correction terms are more precise than those obtained through a GLMM with a gamma distribution, as long as the parameter $\alpha$ associated with shape, in the gamma distribution is not close to one. And, even when the parameter is one, corresponding to an exponential distribution, as the number of clusters and observations in each cluster increase, the estimations obtained using the correction terms are close to those obtained with the GLMM and a gamma distribution (being in general better the ones using the smearing estimate, specially for lower values associated with the variance of the random effect, and viceversa), and better that those obtained through another correction method or without correction terms. On the other hand, in other type of analyses, as in some small area estimation techniques, it is desirable to preserve a normal distribution since the fit of a RIM is just one first step in a set of processes, all assuming normality; hence, assuming another distribution would change the complete technique; and, without the correction, the estimated poverty measures or any measure associated with a small area might be incorrect. The weights we considered for heteroscedasticity were of the form $\sigma^2 w_{ij}^{-1}$; however, a more general form $\sigma_{ij}^2$ can be used by substituting $\sigma_{ij}^2$ for $\sigma^2 w_{ij}^{-1}$ in all formulas. If the variance structure is estimated using a function, for instance an exponential variance structure, we estimate the LMM including this structure. Thus, the parameters for the structure are estimated with the fixed and random effects parameters. Any inference should be performed being careful that the degrees of freedom are corrected or appropriate corrections applied, particularly for small sample sizes [28] and non-linear covariance structures [29]. For the predictions in the original scale, the $\sigma_{ij}^2$ terms can be calculated using the estimated parameters corresponding to the variance structure and then using our formulas. Any further inference should be taken with care considering the variance structure was estimated. In fact, assuming any correlation structure associated with the error for each cluster, i.e. assuming that the matrix $\Sigma_i$ is not necessarily diagonal (however, the correlation structure between clusters is still assumed diagonal), for instance when time is involved, Eqs (15) and (16) still hold true, and formula (17) might be used modifying the third term accordingly, though care should be taken if any inference is required.

We also generalized the procedure considering any LMM, being RIM a particular case, and outlined the process that could be followed when a function different from the logarithm is used, though it seems that approximations should be used in this general case. Future work could be to continue working with transformations different from the logarithm to see if better predictions with closed formulas can be obtained. An exact variance estimator of the predicted values is also something desirable, though it seems, from some preliminary calculations, that a closed formula cannot be obtained; however, a better approximation than one

using only simulations might be possible. We are working in the implementation of the correction terms in two-part models and their variants, for instance for health expenditure data in which there is concentration in the zero value since some people do not spend money, to see whether our correction terms allow to obtain better predictions as some preliminary analyses have shown.

## Supporting information

**S1 File. Proposition 1 and Proposition 2 concerning the calculations to obtain Var($\gamma_i$|log (Y)) and Var($\gamma_i$|log(Y)) for a RIM and LMM, respectively, and details about the mimic simulation example.**
(PDF)

**S2 File. R code associated with the non-simulated data analyzed in the manuscript.**
(R)

**S3 File. Source R code that allow to replicate the analysis for the mimic simulation example.**
(R)

**S1 Data. Data that allow to replicate the analysis for the mimic simulation example.**
(CSV)

**S1 Text. Instructions that allow to replicate the analysis for the mimic simulation example.**
(TXT)

**S2 Text.**
(TXT)

## Acknowledgments

We thank to Dra. Mónica Tinajero Bravo from Consejo Nacional de Evaluación de la Política de Desarrollo Social (CONEVAL), Mexico, for her critical remarks and interesting discussion on the proposal of this paper.

## Author Contributions

**Conceptualization:** Ricardo Ramírez-Aldana.

**Data curation:** Ricardo Ramírez-Aldana, Lizbeth Naranjo.

**Formal analysis:** Ricardo Ramírez-Aldana, Lizbeth Naranjo.

**Investigation:** Ricardo Ramírez-Aldana, Lizbeth Naranjo.

**Methodology:** Ricardo Ramírez-Aldana.

**Software:** Ricardo Ramírez-Aldana, Lizbeth Naranjo.

**Supervision:** Ricardo Ramírez-Aldana.

**Visualization:** Ricardo Ramírez-Aldana, Lizbeth Naranjo.

**Writing – original draft:** Ricardo Ramírez-Aldana.

**Writing – review & editing:** Ricardo Ramírez-Aldana, Lizbeth Naranjo.

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
