## [Decision Letter · Decision Letter 0]

10 Nov 2020

PONE-D-20-31434

Random intercept model including heteroscedasticity in a logarithmic scale: correction terms and prediction in the original scale

PLOS ONE

Dear Dr. Ramírez-Aldana,

Thank you for submitting your manuscript to PLOS ONE. One of the referees have recommended that you make amendments to your article. We therefore invite you to submit a revised version of the manuscript that addresses the points raised during the review process.

We look forward to receiving your revised manuscript.

Kind regards,

Ivan Kryven

Academic Editor

PLOS ONE

Journal Requirements:

Reviewers' comments:

Reviewer's Responses to Questions

**Comments to the Author**

1. Is the manuscript technically sound, and do the data support the conclusions?

Reviewer #1: Yes

Reviewer #2: Yes

2. Has the statistical analysis been performed appropriately and rigorously? 

Reviewer #1: Yes

Reviewer #2: Yes

3. Have the authors made all data underlying the findings in their manuscript fully available?

Reviewer #1: No

Reviewer #2: Yes

4. Is the manuscript presented in an intelligible fashion and written in standard English?

Reviewer #1: Yes

Reviewer #2: Yes

5. Review Comments to the Author

Reviewer #1: This paper proposes and evaluates correction methods for analysis using log-transformed data to fit a linear mixed model and using the fit to obtain predictions on the original scale.

(1) The correction follows from the property of the log-normal distribution. If mu and var are the mean and variance on the log-scale than exp(mu + var/2) is the mean on the original scale. This is well-known, so the claim made in the paper that the correction is proposed "for the first time" is probably overstating the case.

(2) That being said, the authors have a particular application. My feeling is that the whole story line should be more focused on the application case. Most importantly, I am missing a clear statement what exactly is to be estimated in the context of the example. Without that context, it is cumbersome to follow all the equations. Not because of the equations, but because it is unclear what we are estimating!

(3) While I think it is important to focus more clearly on the example, I also think the scope needs to be broadened a bit for the benefit of readers using mixed models but not necessarily the one considered by the authors. The corrections are pretty straight-forward and applicable with other mixed models. All that's required is to work out the relevant marginal variance to use in the correction.

(4) Model (2) only has a single subscript "i", whereas equation (4) has two subscripts "ij". It is not clear what these subscripts refer to and why they are changing in this way. Again, providing the specific context of your example may help here.

(5) In line 169, I am not sure why there is a need to switch to matrix notation. What does this really help? I am also not sure expressions like in eq. (12) are correct, where you are mixing matrix and scalar expressions.

(6) Line 219: What is the rationale underlying the smearing estimator, and what is it's advantage over the plug-in estimator based on what I said in (1)? Some explanation in the paper would be useful.

(7) Line 267: It is important to tell readers how you simulated the data and why.

(8) Line 274: Why did you even consider fitting this GLMM? It's not the model used to simulate the data, is it? Conversely, if it is assumed the GLMM generated the data, why use any other approach for analysis? Clearly, a rationale for using these alternate models is missing.

(9) Line 281: On a similar note, it is not clear to me why data were generated from a GLMM in a second simulation. I think you need a clear justification and story line for this choice.

(10) The manuscript file did not contain and of the diagnostic residual plots, so I was not able to convince myself that the log-normality assumption was satisfied. How do you actually check this when the data is heteroscedastic, as is the case under your model?

(11) And now the obvious question: what to do if you are not so lucky that the log-transformation works well? You mention Box-Cox transformation in the discussion. What if I need to use x^0.3, say? How does your approach generalize? If you can cover that more general case, and also tell people what to do in case their LMM looks different than yours, you can vastly broaden the scope and impact of your paper. As it stands, it's a rather narrow case study.

Reviewer #2: Authors have proposed correction terms to predict the response in the original scale, which are easy to calculate and implement, for a random intercept model (RIM) with or without heteroscedasticity with response in a logarithmic scale. Different estimators of the predicted response are given (some of them already present in the literature)

In addition, simulations and a real dataset are presented in the paper to show the importance of using the correction terms to obtain more accurate predictions.

I have no comment whatsoever. Overall the manuscript is well written and organized. I enjoyed reading the paper and we can say that this paper contributes to the growing literature on statistical modeling.

6. PLOS authors have the option to publish the peer review history of their article (what does this mean?). If published, this will include your full peer review and any attached files.

Reviewer #1: No

Reviewer #2: No

---

## [Author Response · Author response to Decision Letter 0]

16 Dec 2020

Dear Dr. Ramírez-Aldana,

Thank you for submitting your manuscript to PLOS ONE. One of the referees have recommended that you make amendments to your

article. We therefore invite you to submit a revised version of the manuscript that addresses the points raised during the review process.

We look forward to receiving your revised manuscript.

Kind regards,

Ivan Kryven

Academic Editor

PLOS ONE

Response: Thank you very much for the revision. We have modi\ffied the paper attending to the comments and suggestions raised by the two reviewers. A point-by-point response is provided in an additional file included as part of our submitted revision. The modi\fcations have been marked in red color in the new version of the paper.

Reviewer #1: This paper proposes and evaluates correction methods for analysis using log-transformed data to fit a linear mixed model and using the fit to obtain predictions on the original scale.

(1) The correction follows from the property of the log-normal distribution. If mu and var are the mean and variance on the log-scale than exp(mu + var/2) is the mean on the original scale. This is well-known, so the claim made in the paper that the correction is proposed "for the first time" is probably overstating the case.

(2) That being said, the authors have a particular application. My feeling is that the whole story line should be more focused on the application case. Most importantly, I am missing a clear statement what exactly is to be estimated in the context of the example. Without that context, it is cumbersome to follow all the equations. Not because of the equations, but because it is unclear what we are estimating!

(3) While I think it is important to focus more clearly on the example, I also think the scope needs to be broadened a bit for the benefit of readers using mixed models but not necessarily the one considered by the authors. The corrections are pretty straight-forward and applicable with other mixed models. All that's required is to work out the relevant marginal variance to use in the correction.

(4) Model (2) only has a single subscript "i", whereas equation (4) has two subscripts "ij". It is not clear what these subscripts refer to and why they are changing in this way. Again, providing the specific context of your example may help here.

(5) In line 169, I am not sure why there is a need to switch to matrix notation. What does this really help? I am also not sure expressions like in eq. (12) are correct, where you are mixing matrix and scalar expressions.

(6) Line 219: What is the rationale underlying the smearing estimator, and what is it's advantage over the plug-in estimator based on what I said in (1)? Some explanation in the paper would be useful.

(7) Line 267: It is important to tell readers how you simulated the data and why.

(8) Line 274: Why did you even consider fitting this GLMM? It's not the model used to simulate the data, is it? Conversely, if it is assumed the GLMM generated the data, why use any other approach for analysis? Clearly, a rationale for using these alternate models is missing.

(9) Line 281: On a similar note, it is not clear to me why data were generated from a GLMM in a second simulation. I think you need a clear justification and story line for this choice.

(10) The manuscript file did not contain and of the diagnostic residual plots, so I was not able to convince myself that the log-normality assumption was satisfied. How do you actually check this when the data is heteroscedastic, as is the case under your model?

(11) And now the obvious question: what to do if you are not so lucky that the log-transformation works well? You mention Box-Cox transformation in the discussion. What if I need to use x^0.3, say? How does your approach generalize? If you can cover that more general case, and also tell people what to do in case their LMM looks different than yours, you can vastly broaden the scope and impact of your paper. As it stands, it's a rather narrow case study.

Response: Thank you very much for the revision. We really appreciate your time in reviewing the paper. We have modi\ffied the paper attending to the comments and suggestions raised by you. A point-by-point response is provided in an additional file included as part of our submitted revision. The modi\ffications have been marked in red color in the new version of the paper.

Reviewer #2: Authors have proposed correction terms to predict the response in the original scale, which are easy to calculate and implement, for a random intercept model (RIM) with or without heteroscedasticity with response in a logarithmic scale. Different estimators of the predicted response are given (some of them already present in the literature)

In addition, simulations and a real dataset are presented in the paper to show the importance of using the correction terms to obtain more accurate predictions.

I have no comment whatsoever. Overall the manuscript is well written and organized. I enjoyed reading the paper and we can say that this paper contributes to the growing literature on statistical modeling.

Response:Thank you very much for the revision. We really appreaciate your time in reviewing the paper.

---

## [Decision Letter · Decision Letter 1]

23 Dec 2020

PONE-D-20-31434R1

Random intercept and linear mixed models including heteroscedasticity in a logarithmic scale: correction terms and prediction in the original scale

PLOS ONE

Dear Dr. Ramírez-Aldana,

Thank you for revising your manuscript. As you can see Reviewer 1 still has several comments and we therefore invite you to prepare a new version of the manuscript addressing Reviewer's feedback. Following on Reviewer's advise, please condense the description of the key steps of the approach and include more elaborated discussion of  examples and extensions.

We look forward to receiving your revised manuscript.

Kind regards,

Ivan Kryven

Academic Editor

PLOS ONE

Reviewers' comments:

Reviewer's Responses to Questions

**Comments to the Author**

1. If the authors have adequately addressed your comments raised in a previous round of review and you feel that this manuscript is now acceptable for publication, you may indicate that here to bypass the “Comments to the Author” section, enter your conflict of interest statement in the “Confidential to Editor” section, and submit your "Accept" recommendation.

Reviewer #1: (No Response)

2. Is the manuscript technically sound, and do the data support the conclusions?

Reviewer #1: (No Response)

3. Has the statistical analysis been performed appropriately and rigorously? 

Reviewer #1: (No Response)

4. Have the authors made all data underlying the findings in their manuscript fully available?

Reviewer #1: (No Response)

5. Is the manuscript presented in an intelligible fashion and written in standard English?

Reviewer #1: (No Response)

6. Review Comments to the Author

Reviewer #1: The revision is a substantial improvement because the example now plays a more central role throughout, making it easier to assess the relevance of the proposed method.

(1) The proposed method itself is rather straightforward. If we log-transform data and fit a linear mixed model assuming normality, then the analysis implies that the untransformed data have a log-normal distribution. The naive back-transformation of means from the log-scale to the original scale does now provide unbiased estimators of the expected value on the original scale. This is all well known, and it is also known that the mean on the original scale is simple exp(mu)*(exp(sigma^2/2). All the authors are, therefore, proposing is that we should multiply the naive estimator exp(mu_hat) with the improved estimator exp(mu_hat)*(exp(sigma^2_hat/2). This works straightforwardly for a linear mixed model, we just need to work out what the marginal variance is in each case.

It is not clear why the authors are using so much matrix algebra to make this very simple point. The matrix algebra is clearly unnecessary, even if generalizing this to other models. Thus, I see a lot of scope form simplification. Certainly, the matrix algebra does not make things easier in any way as asserted in L202. How could things be possibly any easier than in the scalar multiplication shown in eq. (9)? And that's all it takes!

(2) In the same vein, the authors' proposal to use simulation in case other transformations than the logarithm are used is fine, though not new either. Moreover, this can be said in a single sentence. Then three pages the authors are devoting to this can be shortened substantially, if not removed altogether. Instead, they could focus on analytical results that are available for the power-normal distribution:

Freeman, J., and R. Modarres. 2006. Inverse Box–Cox: The powernormal distribution. Stat. Probab. Lett 76:764–772. doi:10.1016/j.spl.2005.10.036

(3) The authors make a bit point about heteroscedasticity, but the form of heteroscedasticity is very restrictive (L145): they assume that the variance is sigma12/w, where w are known weights. This is, of course, the most favourable case, but most of the time such weights are unknown. This begs two questions: (i) what to do if heteroscadasticity is of a different form and (ii) what are the weights in the central real example? They mention in L32 what the weights could be, but what about their example?

(4) As regards the simulation, it would be very useful to explain how the parameter setting is modelled on the real example. As it stands, it seems like the settings studied fall from the skies, which is not very convincing. In particular, coming back to the weights, what is the rationale for the simulation of the weights in L283-284? What do the weights represent in the real-life application the simulation is hopefully representing here?

Further remarks:

L141: Something wrong here with the representation of the normal as N(sigma^2, sigma^2)

L154: y => and

7. PLOS authors have the option to publish the peer review history of their article (what does this mean?). If published, this will include your full peer review and any attached files.

Reviewer #1: No

---

## [Author Response · Author response to Decision Letter 1]

4 Feb 2021

PONE-D-20-31434R1

Random intercept and linear mixed models including heteroscedasticity in a logarithmic scale: correction terms and prediction in the original scale

PLOS ONE

Dear Dr. Ramírez-Aldana,

Thank you for revising your manuscript. As you can see Reviewer 1 still has several comments and we therefore invite you to prepare a new version of the manuscript addressing Reviewer's feedback. Following on Reviewer's advise, please condense the description of the key steps of the approach and include more elaborated discussion of examples and extensions.

We look forward to receiving your revised manuscript.

Kind regards,

Ivan Kryven

Academic Editor

PLOS ONE

Response: Thank you very much for the revision. We have modified the paper and supplementary material attending to the comments and suggestions raised by Reviewer 1. A point-by-point response was uploaded. The modifications have been marked in red color in the new version of the paper. 

We have condensed the description of the steps of our approach by providing details in the Supplementary Material. We have also included more discussion concerning the example and extensions, condensing all these extensions. Additionally, a mimic example based on the real data set with its R code, data, and instructions to replicate it is now included as part of the manuscript.

It is important to note that since the last revision, we have added funding for this research, and some lines have to be included in the manuscript concerning this matter. We have added them next to the Acknowledgment section.

Review Comments to the Author

Reviewer #1: The revision is a substantial improvement because the example now plays a more central role throughout, making it easier to assess the relevance of the proposed method.

Thank you very much for the revision. We really appreciate your time in reviewing the paper. 

We have modified the paper attending to the comments and suggestions raised by you. A point-by-point response was uploaded. The modifications have been marked in red color in the new version of the paper.

---

## [Decision Letter · Decision Letter 2]

9 Feb 2021

PONE-D-20-31434R2

Random intercept and linear mixed models including heteroscedasticity in a logarithmic scale: correction terms and prediction in the original scale

PLOS ONE

Dear Dr. Ramírez-Aldana,

Thank you for resubmitting your manuscript to PLOS ONE. Reviewer 1 has noticed that one of their comments was not understood as intended, please see below. We therefore invite you to respond to this comment by incorporating the necessary changes to the manuscript.

We look forward to receiving your revised manuscript.

Kind regards,

Ivan Kryven

Academic Editor

PLOS ONE

Reviewers' comments:

Reviewer's Responses to Questions

**Comments to the Author**

1. If the authors have adequately addressed your comments raised in a previous round of review and you feel that this manuscript is now acceptable for publication, you may indicate that here to bypass the “Comments to the Author” section, enter your conflict of interest statement in the “Confidential to Editor” section, and submit your "Accept" recommendation.

Reviewer #1: (No Response)

2. Is the manuscript technically sound, and do the data support the conclusions?

Reviewer #1: (No Response)

3. Has the statistical analysis been performed appropriately and rigorously? 

Reviewer #1: (No Response)

4. Have the authors made all data underlying the findings in their manuscript fully available?

Reviewer #1: (No Response)

5. Is the manuscript presented in an intelligible fashion and written in standard English?

Reviewer #1: (No Response)

6. Review Comments to the Author

Reviewer #1: The authors acknowledge that in case of heteroscedasticity the weights are typically unknown and hence need to be estimated. They further mention two options in R to estimate such functions and note that these involve parameters. It may be added that common forms of heteroscedasticity may also involve correlation, as is the case, e.g., with repeated measures data. The authors further response is based on this assertion: "As far as we know, once these parameters are estimated, the weights could be assumed as known, by simply calculating these values for each individual using the estimated parameters." This is true as far as the point estimates for fixed and random effects are concerned, and only when residual errors are independently distributed, but it is incorrect as far as the inference is concerned. Adjustments are needed both regarding the standard errors and the degrees of freedom. See the papers by Kenward and Roger (1997 and 2009 in Biometrics and CSDA).

I would like the authors to re-consider this point, and I am happy to take another look once that's done.

7. PLOS authors have the option to publish the peer review history of their article (what does this mean?). If published, this will include your full peer review and any attached files.

Reviewer #1: No

---

## [Author Response · Author response to Decision Letter 2]

25 Mar 2021

PONE-D-20-31434R2

Random intercept and linear mixed models including heteroscedasticity in a logarithmic scale: correction terms and prediction in the original scale

PLOS ONE

Dear Dr. Ramírez-Aldana,

Thank you for resubmitting your manuscript to PLOS ONE. Reviewer 1 has noticed that one of their comments was not understood as intended, please see below. We therefore invite you to respond to this comment by incorporating the necessary changes to the manuscript.

We look forward to receiving your revised manuscript.

Kind regards,

Ivan Kryven

Academic Editor

PLOS ONE

Answer: Thank you very much for the revision. We have modified the paper attending to the comment raised by Reviewer 1. A point-by-point response is provided as an additional file. The modifications have been marked in red color in this new version of the paper.

6. Review Comments to the Author

Reviewer #1: The authors acknowledge that in case of heteroscedasticity the weights are typically unknown and hence need to be estimated. They further mention two options in R to estimate such functions and note that these involve parameters. It may be added that common forms of heteroscedasticity may also involve correlation, as is the case, e.g., with repeated measures data. The authors further response is based on this assertion: "As far as we know, once these parameters are estimated, the weights could be assumed as known, by simply calculating these values for each individual using the estimated parameters." This is true as far as the point estimates for fixed and random effects are concerned, and only when residual errors are independently distributed, but it is incorrect as far as the inference is concerned. Adjustments are needed both regarding the standard errors and the degrees of freedom. See the papers by Kenward and Roger (1997 and 2009 in Biometrics and CSDA).

I would like the authors to re-consider this point, and I am happy to take another look once that's done.

Answer: Thank you very much for the revision. We really appreciate your time in reviewing the paper. We have modified the paper attending to the comment raised by you. A full response is provided as an additional file. The modifications have been marked in red color in the new version of the paper.

---

## [Editor Report · Decision Letter 3]

29 Mar 2021

Random intercept and linear mixed models including heteroscedasticity in a logarithmic scale: correction terms and prediction in the original scale

PONE-D-20-31434R3

Dear Dr. Ramírez-Aldana,

We’re pleased to inform you that your manuscript has been judged scientifically suitable for publication and will be formally accepted for publication once it meets all outstanding technical requirements.

Kind regards,

Ivan Kryven

Academic Editor

PLOS ONE
---

## [Editor Report · Acceptance letter]

31 Mar 2021

PONE-D-20-31434R3 

Random intercept and linear mixed models including heteroscedasticity in a logarithmic scale: correction terms and prediction in the original scale 

Dear Dr. Ramírez-Aldana:

I'm pleased to inform you that your manuscript has been deemed suitable for publication in PLOS ONE. Congratulations! Your manuscript is now with our production department. 

Kind regards, 

on behalf of

Dr. Ivan Kryven 

Academic Editor

PLOS ONE